



# Street-in-Grid modeling of gas-phase pollutants in Paris city

Lya Lugon[1,2], Karine Sartelet[1], Youngseob Kim[1], Jérémy Vigneron[3], and Olivier Chrétien[2]

[1]CEREA, Joint Laboratory École des Ponts ParisTech/EDF R&D, Université Paris-Est, 77455 Champs-sur-Marne, France
[2]Paris City, Department of green spaces and environment, 103 Avenue de France, France
[3]Airparif, France

**Correspondence:** Lya Lugon (lya.lugon@enpc.fr), Karine Sartelet (karine.sartelet@enpc.fr)

**Abstract.** Regional-scale chemistry-transport models have coarse spatial resolution, and thus can only simulate background concentrations. They fail to simulate the high concentrations observed close to roads and in streets, i.e. where a large part of the urban population lives. Local-scale models may be used to simulate concentrations in streets. They often assume that background concentrations are constant and/or use simplified chemistry. Recently developed, the multi-scale model Street-in-

Grid (SinG) estimates gaseous pollutant concentrations simultaneously at local and regional scales, coupling them dynamically. This coupling combines the regional-scale chemistry-transport model Polair3D and the street network model MUNICH (Model of Urban Network of Intersecting Canyons and Highway). MUNICH models explicitly street canyons and intersections, and it is coupled to the first vertical level of the chemical-transport model, enabling the transfer of pollutant mass between the street canyon roof and the atmosphere. The original versions of SinG and MUNICH adopt a stationary hypothesis to estimate

pollutant concentrations in streets. Although the computation of $NO_x$ concentration is numerically stable with the stationary approach, the partitioning between NO and $NO_2$ is highly dependent on the time step of coupling between transport and chemistry processes. In this study, a new non-stationary approach is presented with a fine coupling between transport and chemistry, leading to numerically stable partitioning between NO and $NO_2$. Simulations of NO, $NO_2$ and $NO_x$ concentrations over Paris city with SinG, MUNICH and Polair3D are compared to observations at traffic and urban stations to estimate the

added value of multi-scale modeling with a dynamical coupling between the regional and local scales. As expected, the regional chemical-transport model underestimates NO and $NO_2$ concentrations in the streets. However, there is a good agreement between the measurements and the concentrations simulated with MUNICH and SinG. The dynamic coupling between the local and regional scales tends to be important for streets with an intermediate aspect ratio and with high traffic emissions.

## 1 Introduction

Air pollution is a serious problem in many cities due to its considerable impacts on human health and the environment, as reported in World Health Organization (WHO) (2006), Brønnum-Hansen et al. (2018), Lee et al. (2018), Chen et al. (2019), Katoto et al. (2019), De Marco et al. (2019). These impacts motivated the development of air-quality models, that estimate pollutant dispersion at determined spatial scales. These models are largely employed to calculate the population exposure and they can support public strategies for pollution control.





Regional-scale chemistry-transport models (CTMs), as three-dimension gridded Eulerian models (e.g. Polair3D (Sartelet et al., 2007), WRF-Chem (Zhang et al., 2010), CHIMERE (Menut et al., 2014), CMAQ (Community Multi-scale Air Quality Modeling System) (Byun and Ching, 1999), AURORA (Mensink et al., 2001)) solve a chemistry-transport equation for chemical compounds or surrogates, taking into account pollutant emissions, transport (advection by winds, turbulent diffusion), chemical transformations, and dry/wet depositions. The simulated concentrations at each grid cell are averaged over the whole

cell surface, often with resolution coarser than 1 km$^2$. CTMs are largely employed to simulate background concentrations, but they are not able to represent the gradients of concentrations observed between near-traffic areas and background. Indeed, in streets, for several pollutants, the concentrations are considerably higher than background ones, due to the proximity of traffic emissions and reduced natural ventilation. It is the case for $NO_2$, for example, which is emitted by traffic and also formed in the atmosphere. Therefore, many street-network models were formulated specifically in the last decades to estimate pollutant

concentrations at the local scale more accurately, with a relatively low computational cost.

The first street-network models were the STREET model (Johnson et al., 1973) and the Hotchkiss and Harlow model (Hotchkiss and Harlow, 1973). The STREET model uses a very simplified parametrization, where the concentration in a street is assumed to be the sum of a street contribution ($c_s$) generated by traffic emissions and a background contribution ($c_b$). STREET was formulated using empirical parameters based on measurements performed in streets of San Jose and St. Louis.

The Hotchkiss and Harlow model is an analytical street-canyon model. It implements an approximate solution of the steady-state advection-diffusion equation, using an eddy diffusivity formulation to describe pollutant dispersion. However, this model assumes a square-root dependency between pollutant dilution and the distance from the source, which may not be appropriate in street canyons, where source-receptor distances are short (Berkowicz et al., 1997).

Afterward, street-network models, such as CALINE4 (California Line source dispersion model) (Benson, 1984), (Sharma

et al.) and CAR (Calculation of Air pollution from Road traffic model) (Eerens et al., 1993), assume that pollutant dispersion follows a Gaussian plume distribution and traffic emissions are line sources. Other models expanded this formulation combining a Gaussian plume and a box model, e.g. CPBM (Canyon Plume Box Model) (Yamartino and Wiegand, 1986), OSPM (Operational Street Pollution Model) (Berkowicz et al., 1997; Berkowicz, 2000), and ADMS-Urban (Atmospheric Dispersion Modeling System) (McHugh et al., 1997). The Gaussian plume model is used to estimate the direct contribution of traffic

emissions, and the box model calculates the recirculation contribution, resultant from the wind vortex formed in the street canyon.

With a different approach, SIRANE (Soulhac et al., 2011, 2012, 2017) uses a box model to determine pollutant concentrations in street canyons, assuming that concentrations are uniform along each street segment. SIRANE considers horizontal wind advection, mass transfer between streets at street intersections, turbulent vertical transfer between streets and the

free atmosphere. Background concentrations above streets are calculated using a Gaussian plume distribution. The simplified parametrizations for airflow and mass transfer implemented in SIRANE are based on computational fluid dynamic simulations and wind tunnel experiments (Soulhac et al., 2008, 2009). The box model is applied to streets with an aspect ratio $\alpha_r$ higher than 0.3, with $\alpha_r = H/W$, $H$ and $W$ are the street height and width respectively (Landsberg, 1981). If $\alpha_r$ is lower than 0.3, the street is treated as an open terrain, and the concentrations are taken equal to background concentrations above the street,





and they are simulated with a Gaussian plume model. However, estimating background concentrations above streets with a Gaussian plume model inhibits a comprehensive atmospheric chemistry treatment, impacting the modeling of secondary pollutant concentrations, such as $O_3$ and the secondary formation of $NO_2$ concentrations. Although SIRANE uses a stationary hypothesis for pollutant transport, a new version of SIRANE, named SIRANERISK (Soulhac et al., 2016), removes the steady state hypothesis and simulates dispersion above street canyons using a Gaussian puff model.

The Model of Urban Network of Intersecting Canyons and Highways (MUNICH) (Kim et al., 2018) presents a similar box-model parameterization as SIRANE, but it does not employ a Gaussian model to determinate background concentrations. They may be provided by measurements, as in Kim et al. (2018), or regional-scale CTMs, as in our study. This approach allows the implementation of a comprehensive chemical module to better estimate secondary pollutant formation. MUNICH differentiates three types of street canyons: ($i$) narrow canyons with $\alpha_r > 2/3$, ($ii$) intermediate canyons with $1/3 \leq \alpha_r \leq 2/3$,
and wide canyons ($iii$) with $\alpha_r < 1/3$. The aspect ratio $\alpha_r$ is used to determine the wind speed in the streets and the vertical mass transfer between the streets and the atmosphere.

Despite this large diversity of parameterizations increasingly complex, local-scale models often assume that background concentrations are constant and/or use simplified chemistry. Although MUNICH is able to consider the temporal and spatial evolution of background concentrations, the coupling between the background and street concentrations is not dynamic. In
other words, the concentrations calculated in the streets do not influence the background concentrations. The coupling between background and street concentrations is dynamic in the multi-scale Street-In-Grid (SinG) model (Kim et al., 2018), which couples the regional scale model Polair3D (Sartelet et al., 2007) to the street-network model MUNICH, using the Polyphemus platform (Mallet et al., 2007). The street-network model is coupled to the first vertical level of the regional scale model. At each time step, the mass transfer between the street and the atmosphere influences both background and street concentrations.
Thus, SinG combines dynamically an advanced treatment of atmospheric transport and chemistry at the regional scale with a street-network parametrization formulated for streets with different aspect ratios. Kim et al. (2018) validated SinG over a street-network located at a Paris suburb, regarding $NO_2$, $NO$ and $NO_x$ concentrations. Compared to the street or to the regional model, the SinG multi-scale approach improved $NO_2$ and $NO_x$ simulated concentrations compared to observations. However, the original version of MUNICH and SinG assume a stationary hypothesis to calculate pollutant transport in streets. As shown
later in this work, the stationary hypothesis impacts secondary pollutant formation and the concentrations of reactive species, such as $NO_2$.

The dynamic coupling between 3D chemistry-transport and local-scale models started with modeling plumes from tall stacks, as described in Seigneur et al. (1983), Karamchandani et al. (2002), Karamchandani et al. (2006), Morris et al. (2002b) and Morris et al. (2002a). In all these studies a dynamic interaction between local and regional scales is performed: the average
grid concentration is used as background concentration to calculate plume dispersion, and the pollutant concentrations present in the plume are mixed to the grid concentrations depending on the plume characteristics. Different criteria are applied to define the moment where the pollutant concentrations of the plume are mixed to the grid concentrations. The criteria vary with the plume size and the mature plume stage (based on chemical reactions). Karamchandani et al. (2011) present an overview of subgrid scale plume models, also named "Plume-in-Grid" (PinG) models. Over time, PinG models have been generalized to deal





with different types of emission sources, such as linear and surface sources, allowing a more accurate modeling of dispersion
around ship emissions and traffic emissions from roadways (Vijayaraghavan et al., 2006; Freitas et al., 2007; Vijayaraghavan
et al., 2008; Cariolle et al., 2009; Briant and Seigneur, 2013; Rissman et al., 2013).

For streets, several models consider a multi-scale modeling between streets and background concentrations, although this
multi-scale is most often not dynamic. Jensen et al. (2017) performed a high resolution multi-scale air-quality simulation for
all streets in Denmark in 2012 using the model THOR (Brandt et al., 2001c, a, b), which combines three air-quality models at
different spatial scales: DEOM (Danish Eulerian Operational Model), which provides regional background concentrations to
UBM (Urban Background Scale Modeling), which then provides urban background concentrations to OSPM at the local scale.
Comparisons between the annual average concentrations calculated with THOR and measured at air-quality stations show a
fairly good agreement, especially for $NO_2$, whereas $PM_{2.5}$ and $PM_{10}$ are underestimated. With this kind of non-dynamic multi-
scale modeling, traffic emissions are counted twice: they are input to the street model to estimate street concentrations, as well
as to the regional model to estimate background concentrations. To avoid this double counting in multi-scale modeling, Stocker
et al. (2012) used a different approach: the Gaussian model ADMS-Urban is applied to estimate the initial dispersion of traffic
emissions during a mixing time $\tau_m$ (typically 1 hour). The multi-scale concentrations are obtained by subtracting the gridded
concentrations simulated after this mixing time $\tau_m$ to the sum of the local-scale concentrations simulated with ADMS-Urban
and the regional-scale concentrations. Hood et al. (2018) applied this model over London for 2012, using the regional-scale
model EMEP4UK (Vieno et al., 2009), to simulate $NO_2$, $NO_x$, $O_3$, CO, $PM_{2.5}$ and $PM_{10}$ concentrations. They showed that the
multi-scale model improves $NO_2$ and particulate concentrations compared to the regional model, especially at near-road sites.

The objective of this work is to quantify the effect of a dynamic multi-scale modeling between the regional and local scales
on NO, $NO_2$ and $NO_x$ concentrations over the street network of Paris city. To do so, SinG, MUNICH and Polair3D simulated
concentrations are compared. Different aspects related to model hypothesis and numerical parameters are studied: the impact
of the stationary hypothesis often used for pollutant dispersion in streets and the time step stability. Model validation is done
by comparing simulated and observed concentrations at both traffic and urban background stations. The local, regional and
multi-scale models MUNICH, Polair3D and SinG are presented in the first section of this paper. The second section describes
the setup of the simulations over Paris city. The third section studies the impact of the stationary hypothesis and the numerical
stability of the multi-scale model. The fourth section compares the simulated concentrations with air-quality measurements at
traffic and background stations. Finally, the fifth section studies the influence of the dynamic coupling between the regional
and local scales.

## 2   Model description

Street-in-Grid (SinG) is a multi-scale model that acts as an interface between the 3D chemistry-transport model Polair3D and
the street-network model MUNICH (Model of Urban Network of Intersecting Canyons and Highways). MUNICH is coupled
to the first vertical level of Polair3D and the mass transfer between the local and regional scales is computed at each time step.
More details about the dynamic coupling are described in the section 3 of Kim et al. (2018) and in the section 2.3 of this paper.



This dynamic (two ways) coupling presents several advantages compared to a one-way formulation, as: $(i)$ concentrations at the local and regional scales affect each other; $(ii)$ no double counting of emissions is performed; $(iii)$ the chemical and

physical parametrizations used at the local and regional scales are consistent: both scales use the same chemical module and meteorological data. The regional and local-scale model, Polair3D and MUNICH, are now described emphasizing the numerical parameters and assumptions investigated in this study.

## 2.1 Regional scale - Polair3D

Polair3D, as described in Boutahar et al. (2004) and Sartelet et al. (2007), is a 3D Eulerian model which solves numerically the

chemistry-transport equation, considering advection, diffusion, dry and wet deposition processes and chemical transformations. Polair3D was used in many studies to simulate gas and particle concentrations at regional scale at different locations, e.g. Sartelet et al. (2012), Abdallah et al. (2018), including Greater Paris Sartelet et al. (2018), Zhu et al. (2016a), Zhu et al. (2016b), Kim et al. (2015), Kim et al. (2014), Couvidat et al. (2013), Royer et al. (2011).

Polair3D numerically solves the chemistry-transport equation by applying a first-order operator, splitting between transport

and chemistry with the sequence: advection-diffusion-chemistry (Korsakissok et al., 2006). Pourchet et al. (2005) performed divers numerical tests with Polair3D. They showed that pollutant concentrations are not significantly influenced by the splitting method nor the splitting time step.

## 2.2 Local scale - MUNICH

The Model of Urban Network of Intersecting Canyons and Highways (MUNICH) is a street-network box model formulated

to calculate pollutant concentrations in street segments. It is composed of two main components: a street-canyon and an intersection components. A complete description of MUNICH may be found in Kim et al. (2018).

MUNICH assumes that the height and width of each street segment are constant, and that concentrations are uniform within the street segment. The time evolution of the mass $Q$ of pollutants in each street segment may be described by equation (1):

$$\frac{dQ}{dt} = \frac{dQ}{dt}\bigg|_{\text{transp}} + \frac{dQ}{dt}\bigg|_{\text{chem}} \tag{1}$$


$$\frac{dQ}{dt}\bigg|_{\text{transp}} = \underbrace{(Q_{inflow} + Q_{emis})}_{inlet\ flux} - \underbrace{(Q_{outflow} + Q_{vert} + Q_{dep})}_{outlet\ flux}, \tag{2}$$

where $Q_{emis}$ represents the traffic mass emission rate, $Q_{inflow}$ the mass inflow rate at intersections, $Q_{vert}$ the turbulent mass flux rate between the atmosphere and the street, $Q_{outflow}$ the outflow flux, and $Q_{dep}$ the deposition flow; each of this term is detailed in Kim et al. (2018). According to Kim et al. (2018), $Q_{outflow}$ is calculated based on outflow air flux (function




of street dimensions, horizontal wind speed) and street concentrations. $Q_{dep}$ depends on deposition rates, and both terms are calculated following equations (3) and (4):

$$Q_{outflow} = Q_{air}C_{st}; \text{ with } Q_{air} = HWu_{st},\tag{3}$$

where H and W are the street height and width, and $u_{st}$ is the mean air velocity in the street,

$$Q_{dep} = F_{dep}C_{st}\tag{4}$$

where $F_{dep}$ is the deposition flow.

According to the equation (8) of Kim et al. (2018), $Q_{vert}$ is inversely proportional to the aspect ratio $\alpha_r$ of the street. Therefore, the vertical mass transfer is more significant for wide streets than for street canyons. The aspect ratio $\alpha_r$ is also used to determine the wind speed in the streets, as described in equations (9), (10) and (11) of Kim et al. (2018).

MUNICH uses a first order splitting scheme between transport and chemistry to solve equation (1).

In the work of Kim et al. (2018), the splitting time step is fixed (100 s) and the time evolution of the mass of pollutants due to transport is computed at each time step using a stationary hypothesis:

$$\left.\frac{dQ}{dt}\right|_{transp} = 0,\tag{5}$$

which leads to the following expressions for the street concentrations $C_{st}$:

$$C_{st} = \frac{Q_{emis} + Q_{inflow} + \gamma C_{bg}}{\gamma + Q_{air} + F_{dep}},\tag{6}$$

where $\gamma$ defines the transfer rate $Q_{vert}$ between the street and the background concentration $C_{bg}$:

$$Q_{vert} = \gamma(C_{st} - C_{bg}) \text{ with } \gamma = \beta\sigma_w WL\frac{1}{1+\alpha_r}\tag{7}$$

with $\beta$ a constant equal to 0.45, $\sigma_w$ the standard deviation of the vertical wind speed, and $W$ and $L$ the width and length of the street.

The time evolution of the concentrations of pollutants due to chemistry is then computed using the chemical module CB05

(Yarwood et al., 2005).

In this study, a new algorithm is defined to calculate pollutant concentrations in streets without the stationary assumption. The non-stationary calculation of pollutant concentrations in streets solves equation (1) using an explicit trapezoidal rule of order 2 (ETR), as in Sartelet et al. (2006). The choice of the initial time step and the time-step adjustment during the simulations are done depending on the evolution of the concentrations due to transport-related processes:

$$\begin{aligned} C^{n+1} &= C^n + \frac{\Delta t}{2}[F(C^n)+F(C^*)]\\ C^* &= C^n + \Delta t\, F(C^n)\end{aligned}\tag{8}$$





where $C^n$ is the concentration at time $t^n$, $F(C^n)$ represents the time derivative of $C^n$ due to transport-related processes and is obtained by equation (2). After each time step $\Delta t$, the time step is adjusted:

$$\Delta t = \Delta t \frac{\Delta_0}{\Delta_1} \text{ with } \Delta_1 = \left\| \left\| \frac{C^{n+1} - C^*}{C^*} \right\| \right\|_2. \tag{9}$$

Because chemical reactions are represented by a stiff set of equations with fast radical chemistry, chemistry processes are solved after transport processes over the time step defined by the ETR algorithm. Note that as in the regional-scale model, chemistry processes are solved with the Rosenbrock algorithm (Voss and Khaliq, 2001) using time steps that may be smaller than the splitting time step defined by the ETR algorithm.

## 2.3    Street-in-Grid model (SinG)

SinG interconnects regional and local scales at each time step. Pollutant concentrations are calculated in streets at the local scale, and they are transferred to the regional scale with a vertical mass flux (see equation (7)) between the street and the regional background concentrations of the first vertical grid level of the CTM. The vertical mass flux corresponds to an emission term for the regional-scale model, and it is used in the local-scale model to compute the time evolution of street concentrations as details in equation (2).

Note that the background concentrations used in equation (7) to compute the vertical mass flux are not exactly those computed by the regional-scale model. Because it does not consider buildings, the volume of the cell in which the concentrations are computed with the regional-scale model is actually larger than the volume of the cell if buildings are considered. Therefore, for each cell $i$, the background concentration over the canopy $C_{bg,cor}^i$ are obtained from regional-scale concentrations corrected to take into account the presence of buildings:

$$C_{bg,cor}^i = \frac{V_{cell}^i}{(V_{cell}^i - V_{build}^i)} C_{bg}^i, \tag{10}$$

where $V_{build}^i$ is the buildings volume, $V_{cell}^i$ is the grid cell volume, and $C_{bg}^i$ is the background concentration calculated over the whole cell volume $V_{cell}^i$ with the regional-scale model.

At each grid cell $i$, SinG performs an average between the pollutant mass in streets ($Q_{st}^i$) and the background pollutant mass ($Q_{bg}^i$) to calculate output concentrations at the regional scale ($C_{reg}^i$), as:

$$C_{reg}^i = \frac{Q_{st}^i + Q_{bg}^i}{V_{cell}} \text{ with } Q_{st} = \sum_{st \ in \ the \ cell} C_{st}^i V_{st} \text{ and } Q_{bg}^i = C_{bg}^i V_{cell}. \tag{11}$$

## 2.4    Setup of air-quality simulations over Paris city

This sections describes the model configuration as well as the input data used for the regional and local-scale simulations. All simulations are performed from the $1^{st}$ to $28^{th}$ May 2014 .





## 2.5  Setup for regional-scale simulations

SinG is applied over Paris city, using a spatial resolution of 1 km × 1 km. Initial and boundary conditions are obtained from one-way nesting simulations using Polair3D over three additional simulations covering Europe (domain 1), France (domain 2) and Île-de-France region (domain 3). The spatial resolution for those simulations is 45 km × 45 km, 9 km × 9 km and 3 km × 3 km, respectively. Figure 1 illustrates the different domains, with domain 4 corresponding to the Paris city domain. The four nested simulations over the domains shown in Figure 1 use the same vertical discretization with 14 levels between 0 and 12000 m, represented in Figure 2.

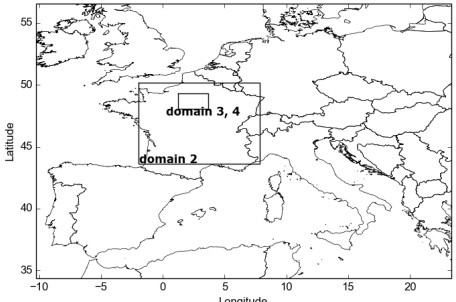

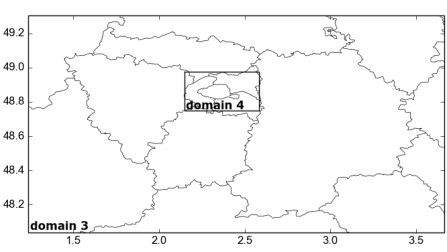

**Figure 1.** Domains simulated: Europe (domain 1), France (domain 2), Île de France region (domain 3), and Paris city (domain 4).


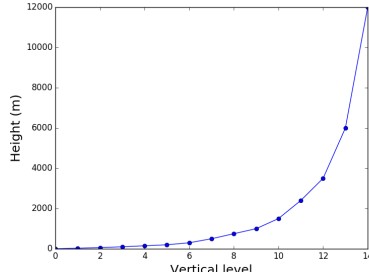

**Figure 2.** Vertical levels employed in all regional-scale simulations.

The initial and boundary conditions of the largest domain (over Europe) are obtained from a global-scale chemical-transport simulation using MOZART-4 (model for Ozone and Related Chemical Tracers) (Emmons et al., 2010) coupled to the aerosol module GEOS-5 (Goddard Earth Observing System Model) (Chin et al., 2002). The spatial resolution of the MOZART-4/GEOS-5 simulation is $1.9° \times 2.5°$, with 56 vertical levels.



Meteorological data for the four domains are calculated by the WRF model (Weather Research and Forecasting) (Skamarock et al., 2008) with a two-way nesting, employing the same spatial resolutions as used in Polair3D nesting simulations (45 km × 45 km, 9 km × 9 km, 3 km × 3 km and 1 km × 1 km for domains 4 to 1 respectively), with 38 vertical levels, from 0 to 5000 m. Observational data of wind speed, wind direction, pressure and temperature from Paris Orly meteorological station are used as input data for the simulations over Paris city (domain 4) using the nudging point technique. WRF domains are represented in

Figure 3, and Table 1 indicates the main physical and chemical options employed in WRF simulations.

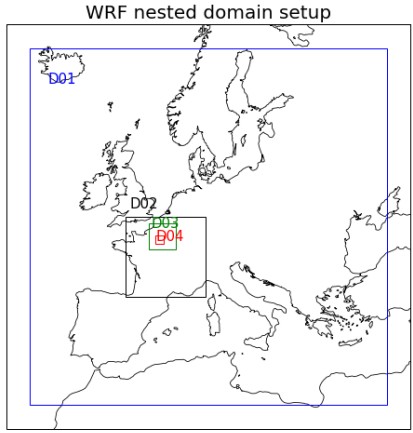

**Figure 3.** Domains simulated using WRF: Europe (D01), France (D02), Île-de-France region (D03), and Paris city (D04).

**Table 1.** Main physical options used in WRF simulations

| | | |
|---|---|---|
| mp_physics | microphysics | WSM 6-class graupel scheme |
| cu_physics | cumulus | Kain-Fritsch (new Eta) scheme |
| ra_lw_physics | longwave radiation | RRTM scheme: Rapid Radiative Transfer Model |
| ra_sw_physics | shortwave radiation | Dudhia scheme |
| bl_pbl_physics | boundary-layer | MYNN 2.5 level TKE scheme |
| sf_sfclay_physics | surface-layer | MYNNSFC |
| sf_surface_physics | land-surface | Noah Land-Surface Model |

Dry-deposition velocities of gas species are estimated following Zhang et al. (2003), and below-cloud scavenging following Sportisse and Du Bois (2002), see Sartelet et al. (2007) for more details on the deposition schemes used. Biogenic emissions over all domains are estimated using the Model of Emissions of Gases and Aerosols from Nature (MEGAN). Concerning anthropogenic emissions, over the domains 1, 2 and outside Île-de-France over the domain 3, they are calculated using EMEP

(European Monitoring and Evaluation Program) emission inventory for the year 2014, with a spatial resolution of 0.1° × 0.1°.





Over Île-de-France of the domain 3 and over the domain 4, they are calculated using the emission inventory of 2012, provided by the air-quality agency of Paris (AIRPARIF). For traffic emissions, AIRPARIF used the HEAVEN bottom-up traffic emissions model (https://trimis.ec.europa.eu/sites/default/files/project/documents/20090917_162316_73833_HEAVEN%20-%20Final%20Report.pdf) with fleet and technology data specific of 2013

and 2014. Anthropogenic emissions followed the vertical distribution defined by Bieser et al. (2011) for the different activity sectors. More details on emission data and speciations may be found in Sartelet et al. (2018).

Note that in SinG, traffic emissions are only considered at the local scale and not at the regional scale to avoid double counting emissions, as shown in Figure 4.

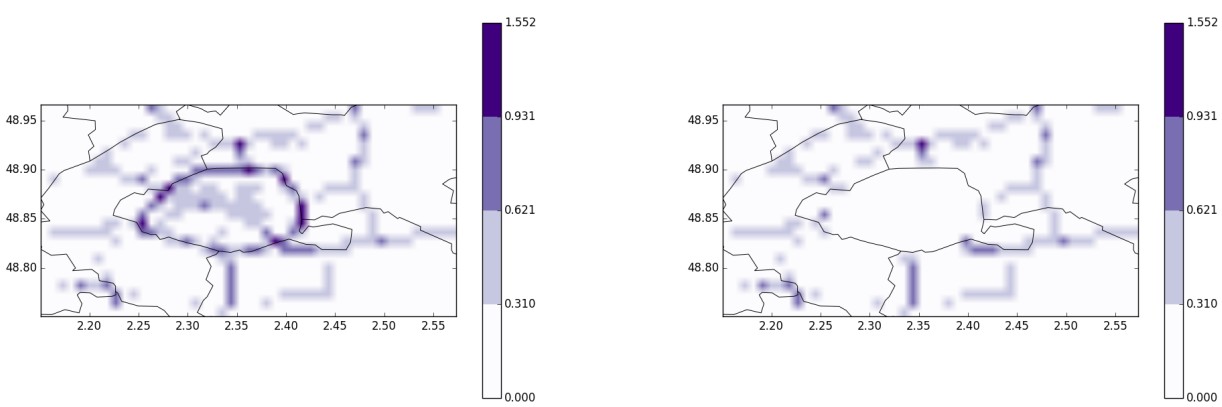

**Figure 4.** For the Paris simulations using Polair3D and SinG, average anthropogenic emissions of NO$_2$ [$\mu$g.s$^{-1}$.m$^{-2}$] used as input of the regional-scale simulation with Polair3D (left panel), and as input of the regional-scale module of the multi-scale simulation with SinG (right panel).

## 2.6 Setup for local-scale simulations

The street network used in this study was provided by AIRPARIF. It contains the main streets of Paris city, totalizing 3819 streets. Apart from the location and length of the street segments, the streets' average dimensions (height and width) need to be defined.

A processing tool was developed to treat three different databases to determine street dimensions. The streets' widths are computed by summing the pavement width (from the BDTOPO database, available at http://professionnels.ign.fr/bdtopo) and

the two sidewalk widths (from an opensource public database "opendataparis", available at https://www.data.gouv.fr/fr/datasets/trottoirs-des-rues-de-paris-prs/). The streets' heights are determined using the Parisian urban planning agency (APUR) database (https://www.apur.org/fr). The average height adopted at each street is calculated considering the mean height of all buildings located near the street axis, with a maximal distance of 10 m.





For the validity of the MUNICH model, buildings' heights cannot be higher than the first vertical level of the regional model,
so a maximum height of 30 m is adopted in this study. This limitation is acceptable over Paris, because the average height of
buildings is about 15 m. A minimum street width equal to 10 m is adopted over the whole domain, imposing 10 m width to
very narrow streets.

A few street segments in the domain, especially along the ring road around Paris ("boulevard périphérique") are tunnels.
For those segments, traffic emissions are not assigned to the segment itself, but to two "virtual" streets added at each tunnel
extremity, with half of the tunnel emissions each. The width of these virtual streets is the same as the width of the tunnel, and
an arbitrary length of 3 m is chosen.

As Paris has an important number of public parks and gardens, the average vegetation height is also considered for streets
along these areas, and the model considers that the street's height is the average height of buildings and trees. The aver-
age trees' height is estimated to be about 13 m, considering the whole domain. It is calculated using a database containing
the height of all trees in public spaces of Paris, available online "opendataparis" (https://opendata.paris.fr/explore/dataset/les-
arbres/information/).

The street network and the street characteristics are used for the local-scale simulations using MUNICH and SinG, where
wind profile and turbulent exchange depend on the aspect ratio $\alpha_r$ (as mentioned in section 2.2) of the streets.

Emission data over the street segments is provided by AIRPARIF using the HEAVEN model (see Sartelet et al. (2018)).
Figure 5 illustrates the average emissions of $NO_2$ during all simulation period. The most important emissions are located along
the ring road ("boulevard périphérique"), as expected. This zone presents the most important road traffic in Paris city.

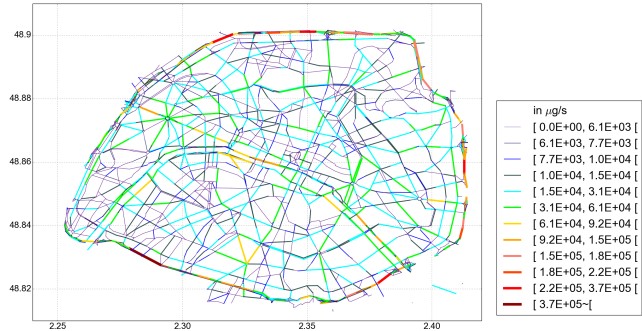

**Figure 5.** Average traffic emissions of $NO_2$ [$\mu g.s^{-1}$] calculated for local-scale simulations

Meteorological data for each street and intersection are obtained from the WRF simulations, as in the regional-scale simu-
lation over Paris city. MUNICH simulations also require background concentrations as input data. They are obtained from a
Polair3D simulation over the Paris city regional-scale domain. Note that the Polair3D simulation uses all emissions, including
traffic, as input data (as indicated in Figure 4).





## 2.7 List of simulations

Different numerical simulations are performed in order to compare the concentrations computed by SinG and MUNICH, as listed bellow. Numerical parameters (main time step) and model hypothesis (stationary hypothesis or not) are analyzed. The main time step corresponds to the splitting time step between transport and chemistry in the regional-scale chemistry-transport

model Polair3D. As in Polair3D, in MUNICH and SinG, the main time step corresponds to the time step used to split local-scale transport and chemistry if the stationary hypothesis is used. If the stationary hypothesis is not made, then the splitting time step between local-scale transport and chemistry is estimated and adjusted as detailed in section 2.2. In SinG, the main time step also corresponds to the splitting time step between the regional-scale (Polair3D) and local-scale (MUNICH) modules. Different simulations are conducted with a main time step equal to 100 s or 600 s, and with or without the stationary hypothesis

in MUNICH and SinG, as detailed in Table 2.

| Sim. number | Model | time step | Stat. hyp. |
|---|---|---|---|
| 1 | MUNICH | 600 s | yes |
| 2 | MUNICH | 100 s | yes |
| 3 | MUNICH | 600 s | no |
| 4 | MUNICH | 100 s | no |
| 5 | SinG | 600 s | yes |
| 6 | SinG | 100 s | yes |
| 7 | SinG | 600 s | no |
| 8 | SinG | 100 s | no |

**Table 2.** List of the sensitivity simulations performed.

Simulated concentrations are compared with air-quality measurements at traffic and urban background stations. Figure 6 represents the street network used in this study, displaying the regional-scale grid mesh and the position of all stations considered. Air-quality stations comprise 5 urban stations (indicated by PA04C, PA07, PA12, PA13 PA18), and 8 traffic stations (BONAP, ELYS, HAUSS, CELES, BASCH, OPERA, SOULT and BP_EST).

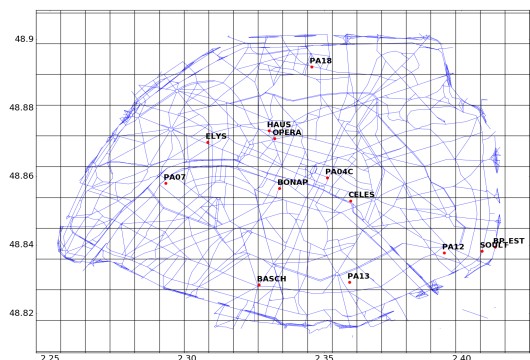

**Figure 6.** Street network with the regional-scale grid mesh and the position of the measurement stations.





## 3 Numerical stability and influence of the stationary hypothesis

As mentioned in section 2.7, different simulations with MUNICH and SinG are performed with different time steps, considering or not the stationary hypothesis. Figures 7 and 8 represent the time evolution of average daily concentrations of $NO_x$, $NO_2$ and NO during the simulation period, as simulated with MUNICH and SinG, at BONAP and CELES stations respectively. $NO_x$ concentrations are independent of whether the stationary hypothesis is made or not, and of the choice of the main time step. However, concentrations of $NO_2$ and NO are highly dependent on the choice of the time step when the stationary hypothesis is made. This time-step dependency is observed using both MUNICH and SinG. This problem is solved with the non-stationary simulations, where concentrations of $NO_2$ and NO are numerically stable and independent of the choice of the main time step. Besides the numerical stability, $NO_2$ and NO average concentrations obtained using the non-stationary approach are closer to observations than those using the stationary hypothesis, as indicated in Table 3. Therefore, in the rest of this paper only the simulations performed with the non-stationary approach and a main time step of 100 s are analyzed.

**Table 3.** Average concentrations (in $\mu$g.m$^{-3}$) observed and calculated with SinG (time-step 100s) using the stationary and non-stationary approaches at CELES (left panel) and BONAP (right panel).

|        | Observations | SinG non-stat. | SinG stat. |
|--------|--------------|----------------|------------|
| $NO_2$ | 55.80        | 64.03          | 85.59      |
| NO     | 49.58        | 51.57          | 37.46      |

(a) CELES

|        | Observations | SinG non-stat. | SinG stat. |
|--------|--------------|----------------|------------|
| $NO_2$ | 46.24        | 54.34          | 61.14      |
| NO     | 43.76        | 25.00          | 20.62      |

(b) BONAP

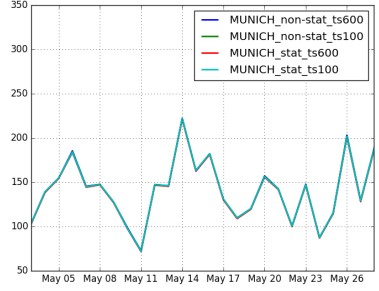
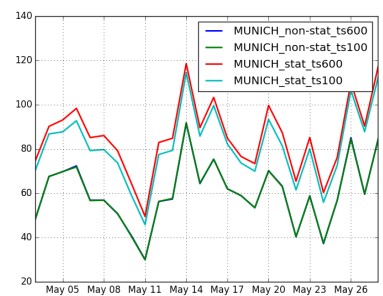
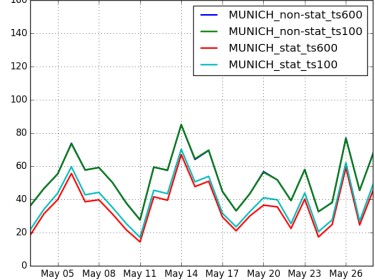

**Figure 7.** Daily-average concentrations of $NO_x$ (left panel), $NO_2$ (middle panel), and NO (right panel) concentrations calculated by MUNICH at CELES station with different main time steps, using the stationary and non-stationary approaches.



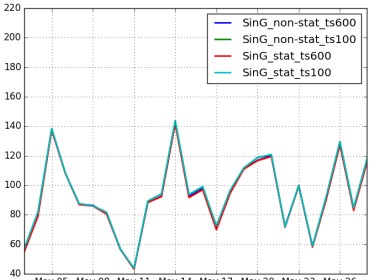 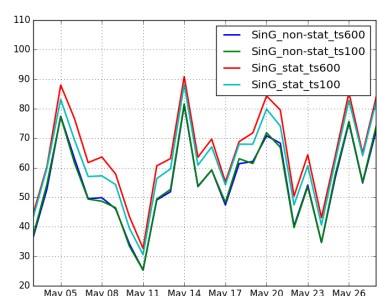 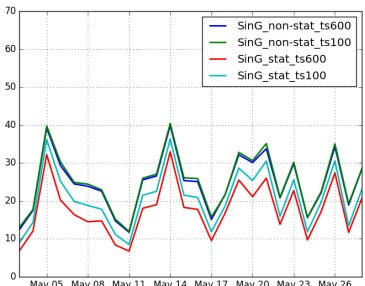

**Figure 8.** Daily-average concentrations of $NO_x$ (left panel), $NO_2$ (middle panel), and NO (right panel) concentrations calculated by SinG at BONAP station with different main time steps, using the stationary and non-stationary approaches.

## 4  Comparisons to air-quality measurements

This section presents the comparisons between the measured concentrations of NO, $NO_2$ and $NO_x$ and those simulated with MUNICH, Polair3D and SinG. As mentioned in section 2.7, air-quality stations comprise eight traffic stations and five urban stations. The criteria applied to evaluate the comparisons are the statistics detailed in Hanna and Chang (2012) and Herring and Huq (2018): $-0.3 < FB < 0.3$; $0.7 < MG < 1.3$; $NMSE < 3$; $VG < 1.6$; $FAC2 \geq 0.5$; $NAD < 0.3$. Hanna and Chang (2012) and Herring and Huq (2018) also defined a less strict criteria to be applied to urban areas: $-0.67 < FB < 0.67$; $NMSE < 6$; $FAC2 \geq 0.3$; $NAD < 0.5$. The definitions of these statistics are given in Annexe A1.

The statistics of the 3 models (Polair3D, MUNICH, SinG) for $NO_2$ and $NO_x$ at traffic and background stations are indicated in Tables 4 and 5 respectively.

**Table 4.** Statistics at traffic stations ($o$ and $s$ represent the average observed and simulated concentrations respectively).

|  | **$NO_2$** | | | | | | | | **$NO_x$** | | | | | | | |
|---|---|---|---|---|---|---|---|---|---|---|---|---|---|---|---|---|
|  | $o$ | $s$ | FB | MG | NMSE | VG | FAC2 | NAD | $o$ | $s$ | FB | MG | NMSE | VG | FAC2 | NAD |
| Polair3D | 59.1 | 21.9 | -0.88 | 0.39 | 1.26 | 3.21 | 0.20 | 0.44 | 146.4 | 27.7 | -1.30 | 0.22 | 4.16 | 33.18 | 0.06 | 0.64 |
| MUNICH | 59.1 | 55.2 | -0.06 | 0.97 | 0.12 | 1.15 | 0.94 | 0.14 | 146.4 | 108.8 | -0.28 | 0.83 | 0.34 | 1.48 | 0.75 | 0.22 |
| SinG | 59.1 | 57.7 | -0.01 | 1.02 | 0.11 | 1.14 | 0.94 | 0.13 | 146.4 | 109.5 | -0.26 | 0.84 | 0.33 | 1.48 | 0.74 | 0.22 |

**Table 5.** Statistics at background stations ($o$ and $s$ represent the average observed and simulated concentrations respectively).

|  | **$NO_2$** | | | | | | | | **$NO_x$** | | | | | | | |
|---|---|---|---|---|---|---|---|---|---|---|---|---|---|---|---|---|
|  | $o$ | $s$ | FB | MG | NMSE | VG | FAC2 | NAD | $o$ | $s$ | FB | MG | NMSE | VG | FAC2 | NAD |
| Polair3D | 31.0 | 21.2 | -0.38 | 0.70 | 0.23 | 1.23 | 0.80 | 0.20 | 38.7 | 28.1 | -0.37 | 0.72 | 0.26 | 1.23 | 0.81 | 0.20 |
| SinG | 31.0 | 23.3 | -0.29 | 0.77 | 0.16 | 1.16 | 0.85 | 0.16 | 38.7 | 30.3 | -0.25 | 0.82 | 0.17 | 1.15 | 0.83 | 0.15 |





## 4.1 Traffic stations

As expected, Polair3D strongly underestimates $NO_2$ and $NO_x$ concentrations at traffic stations, as shown by the statistical indicators of Table 4, and the performance criteria are not respected. However, $NO_2$ and $NO_x$ concentrations are well modeled using both MUNICH and SinG.

As shown in Table 4, both MUNICH and SinG present similar statistics at the local scale, respecting the most strict performance criteria determined by Hanna and Chang (2012) for $NO_2$ and $NO_x$. Compared to MUNICH, the multi-scale approach of SinG improves the average statistical parameters for both pollutants.

The statistics at each station (see Annexe A2) show that the less strict criteria of Hanna and Chang (2012) indicated for urban areas are satisfied at all stations for $NO_2$ concentrations using MUNICH and SinG. The most strict criteria are even respected at all stations except BASCH. In both MUNICH and SinG simulations, NO concentrations tend to be underestimated, although the performance criteria are verified at 6 out of 8 stations. This underestimation may be due to the short life time of NO, leading to high uncertainties on dispersion, and questioning the assumption of uniform concentrations in streets. The NO underestimation is the most significant at stations located in big squares (OPERA and BASCH), indicating that the air flow parameterization for big squares may need to be improved. Note that because of the underestimation of NO concentrations at OPERA and BASCH, the performance criteria for $NO_x$ are not respected at BASCH and only the less strict performance criteria are respected at OPERA.

The daily evolution of $NO_x$, $NO_2$ and NO concentrations is well simulated, as shown in Figures 9 and 10, which display the time evolution of daily concentrations of $NO_x$, $NO_2$ and NO simulated with MUNICH, SinG and Polair3D at CELES and SOULT stations. However, $NO_2$ concentrations are overestimated at almost all stations from the $9^{th}$ to the $11^{th}$ May. This period corresponds to a french holiday, suggesting that the temporal variability of emissions needs to be modified in the model for those days. Beyond daily average concentrations, both SinG and MUNICH represent well the time evolution of hourly concentrations, as shown in Figure 11.

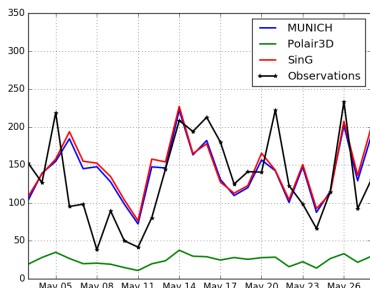
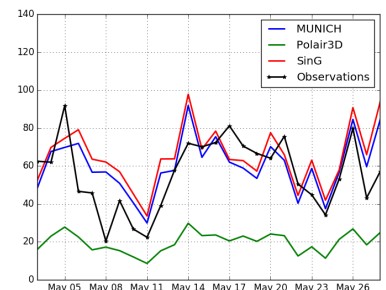
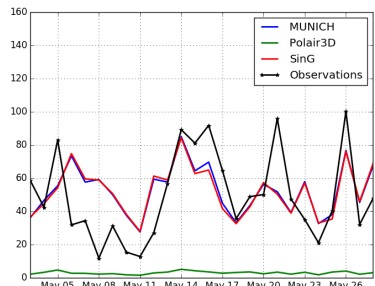

**Figure 9.** Daily-average $NO_x$ (left panel), $NO_2$ (middle panel) and NO (right panel) concentrations observed and simulated at CELES station with MUNICH, SinG and Polair3D



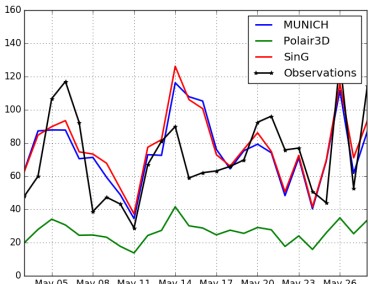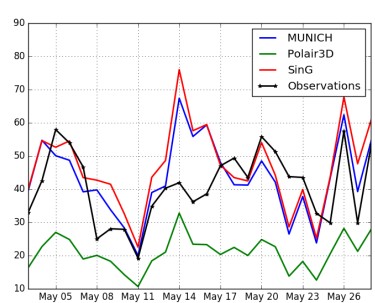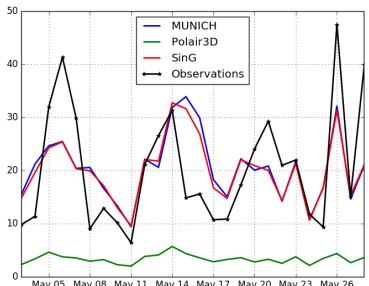

**Figure 10.** Daily-average NO$_x$ (left panel), NO$_2$ (middle panel) and NO (right panel) concentrations observed and simulated at SOULT station with MUNICH, SinG and Polair3D

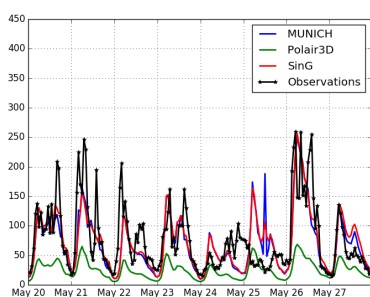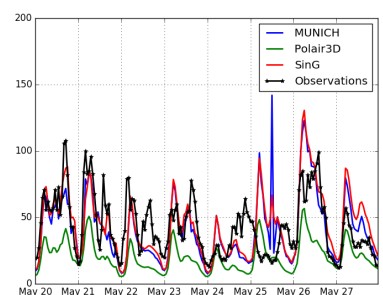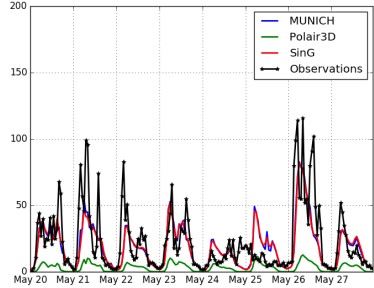

**Figure 11.** Hourly-average NO$_x$ (left panel), NO$_2$ (middle panel) and NO (right panel) concentrations observed and simulated at SOULT station with MUNICH, SinG and Polair3D.

Table 6 indicates the average values of air-quality measurements and SinG concentrations, and the corresponding ratios of NO$_2$/NO. The ratios are overestimated in the simulations: they vary between 0.80 and 2.06 in the measurements, and between 0.98 and 2.80 in the simulations. The ratios are well simulated at CELES, SOULT and BP_EST stations, which are located in streets with high traffic emissions. However, they are strongly overestimated at other stations, such as those in big squares (OPERA, BASCH). This may be due to the short life time of NO, for which the assumption of uniform concentrations in wide streets and big squares may not be verified.





**Table 6.** Average concentrations measured and simulated with SinG of $NO_x$, $NO_2$, NO and $NO_2$/NO ratios at traffic stations ($o$ and $s$ represent the observed and simulated average respectively).

|  | $NO_2$ | | NO | | $NO_x$ | | $NO_2$/NO | |
|---|---|---|---|---|---|---|---|---|
|  | $o$ | $s$ | $o$ | $s$ | $o$ | $s$ | $o$ | $s$ |
| CELES | 55.8 | 64.0 | 49.6 | 51.6 | 131.5 | 143.1 | 1.12 | 1.24 |
| BONAP | 46.2 | 54.3 | 43.7 | 25.0 | 113.1 | 92.7 | 1.06 | 2.17 |
| SOULT | 40.4 | 46.1 | 19.6 | 20.1 | 70.3 | 77.0 | 2.06 | 2.29 |
| ELYS | 51.0 | 49.8 | 38.4 | 18.5 | 109.8 | 78.1 | 1.33 | 2.69 |
| OPERA | 74.3 | 60.3 | 81.1 | 27.7 | 198.5 | 102.8 | 0.92 | 2.17 |
| HAUS | 56.1 | 55.5 | 37.2 | 19.8 | 112.8 | 86.0 | 1.51 | 2.80 |
| BP_EST | 70.8 | 80.3 | 88.6 | 81.5 | 206.3 | 205.2 | 0.80 | 0.98 |
| BASCH | 78.4 | 51.5 | 98.1 | 25.7 | 228.9 | 90.9 | 0.80 | 2.00 |

## 4.2 Background stations

Although both SinG and Polair3D perform well at simulating background $NO_2$ and $NO_x$ concentrations, the multi-scale approach SinG improves the statistics of comparisons to measurements at urban background stations. Table 5 presents the statistics at urban background stations for the $NO_2$ and $NO_x$ concentrations simulated with Polair3D and SinG. The multi-scale approach used in SinG improved all statistical parameters, especially the fractional bias, for both $NO_2$ and NOx. Regarding the simulated period, SinG respects the most strict performance criteria defined by Hanna and Chang (2012).

As expected, the differences between $NO_x$ concentrations simulated with SinG and Polair3D are the highest at stations where vehicular traffic is high. Figures 12 and 13 show the time-evolution of daily NO, $NO_2$ and $NO_x$ concentrations at the background stations PA04C and PA13. PA04C is a station located nearby an important traffic area, while PA13 is located in an area with lower vehicle flux. SinG and Polair3D differences are more important at PA04C station than at PA13 station. More details about the differences of Polair3D and SinG concentrations are described in section 5.2.

Even though both SinG and Polair3D represent both well the measured background concentrations, the dynamic coupling between spatial scales in SinG improves the modelling of $NO_2$, NO and $NO_x$ background concentrations. Furthermore, SinG proved to represent well $NO_2$ and $NO_x$ concentrations both at local (traffic stations) and regional (background stations) scales.





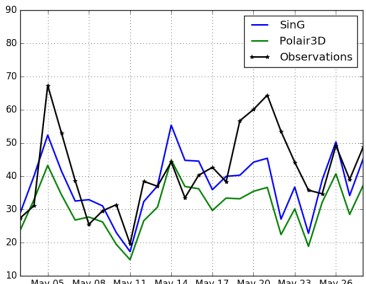 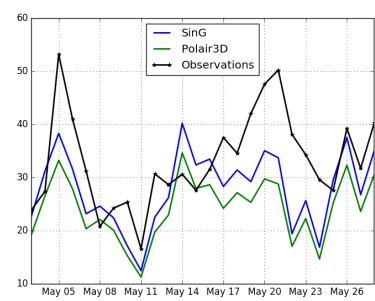 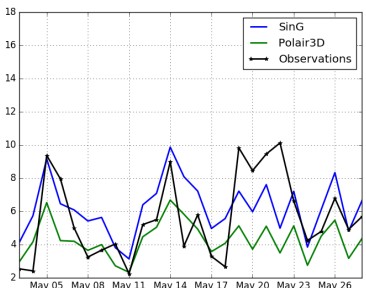

**Figure 12.** Daily concentrations of $NO_x$ (left panel), $NO_2$ (middle panel) and NO (right panel) observed and simulated at PA04C station with SinG and Polair3D.

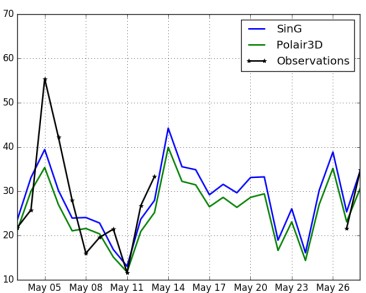 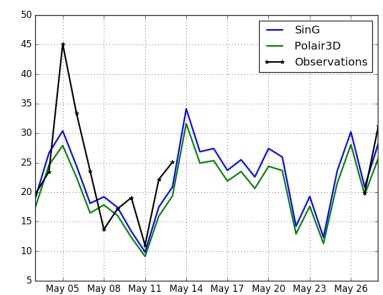 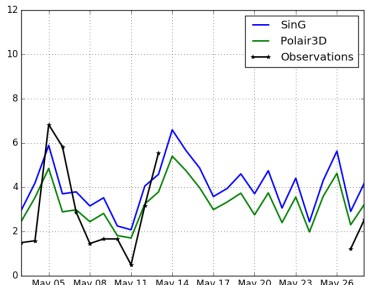

**Figure 13.** Daily concentrations of $NO_x$ (left panel), $NO_2$ (middle panel) and NO (right panel) observed and simulated at PA13 station with SinG and Polair3D.

## 5 Influence of the dynamic coupling between the regional and local scales

This section analyzes the influence of the dynamic coupling between the regional and local scales on NO, $NO_2$ and $NO_x$ concentrations. This influence is analyzed by comparing the concentrations simulated with SinG and MUNICH at the local

scale (in streets), and SinG and Polair3D at the regional scale (background concentrations). The influence of different factors influencing this coupling is evaluated: the geometric characteristics of the streets, the inlet and output mass fluxes in the streets and the intensity of traffic emissions.

At both the regional and local scales, the larger differences between coupled and non-coupled simulations are observed in high traffic emissions areas. In these areas the vertical mass transfer between the local and regional scales tend to be more

important for two main reasons: ($i$) the gradient between street and background concentrations is larger when traffic emissions are higher (see equation 7), and ($ii$) higher traffic emissions lead to higher influence of the mass advection between streets by mean wind, and therefore higher influence of vertical mass transfer at street intersections. Figure 14 represents the mean



relative differences between $NO_2$ concentrations simulated using coupled and non-coupled simulations at local (differences between SinG and MUNICH) and regional scales (differences between SinG and Polair3D), averaged over the simulation

period. In average, these mean relative differences are about 7.5% at the local scale and 11.3% at the regional scale.

The influence of dynamic coupling is now studied in more details, first at the local scale (in streets), and then at the regional scale.

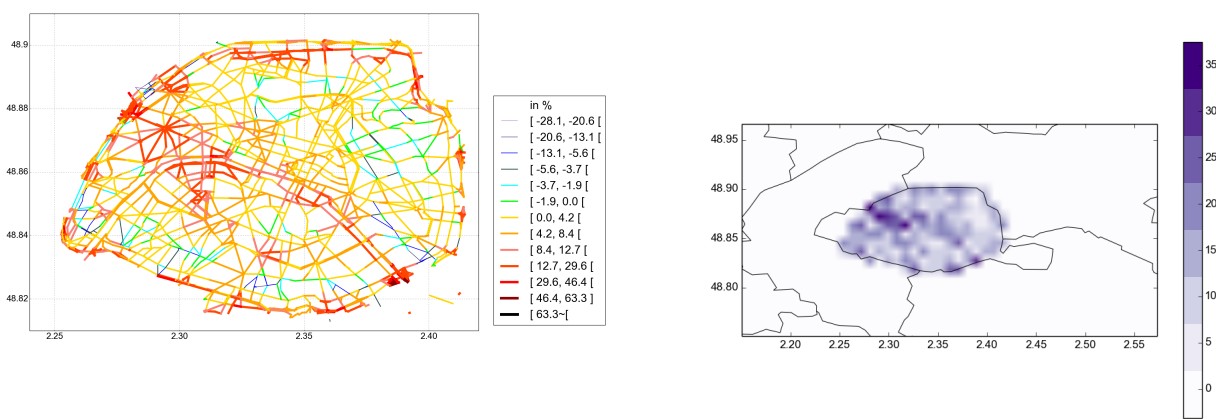

**Figure 14.** Relative differences (in %) between $NO_2$ concentrations simulated by SinG and MUNICH at the local scale (left panel) and by SinG and Polair3D at the regional scale (right panel).

## 5.1   Local scale

The differences between SinG and MUNICH are first analyzed at traffic stations. In SinG, the coupling depends on the con-
centration gradients between the street and the background, but also on the street dimensions, the standard deviation of vertical wind speed, and input/output mass fluxes at intersections. Table 7 summarizes the street characteristics, with $L$ the street length, $\alpha_r$ the street aspect ratio, and $NO_2$ diff$(\%)_{s,m}$ the mean relative difference between $NO_2$ concentrations simulated with SinG and MUNICH over the simulation period. The differences between SinG and MUNICH concentrations are quite low: they are lower than 12% at each of the 8 traffic stations. In agreement with section 4.1 and Table 4, $NO_2$ concentrations simulated with
SinG tend to be larger than those simulated with MUNICH, because the background concentrations in SinG are influenced by the high $NO_x$ concentrations of the street network.

As explained in section 2.3, SinG transfers the vertical mass flux from streets and intersections to the regional scale to correct background concentrations. Therefore, the differences between MUNICH and SinG simulations are mostly due to differences in background concentrations. The time variations of the differences are illustrated in Figure 15, which represents
the time evolution at CELES station of $NO_2$ concentrations in the streets and the background using MUNICH and SinG. The differences between the street and the background concentrations are strongly correlated. Higher are the differences between



**Table 7.** Street characteristics at traffic stations.

| Station | $L$ (m) | $\alpha_r$ | Connec. streets | $NO_2$ diff(%)$_{s,m}$ |
|---------|---------|------------|-----------------|------------------------|
| CELES   | 75.87   | 0.398      | 4               | 10.30                  |
| BONAP   | 267.96  | 1.500      | 3               | 2.81                   |
| SOULT   | 177.51  | 0.498      | 5               | 10.03                  |
| ELYS    | 391.07  | 0.308      | 8               | 11.22                  |
| OPERA   | 315.12  | 0.681      | 5               | 7.68                   |
| HAUS    | 315.03  | 0.860      | 7               | 7.95                   |
| BP_EST  | 362.28  | 0.125      | 3               | -0.46                  |
| BASCH   | 382.74  | 0.463      | 6               | 4.38                   |

SinG and MUNICH background concentrations, higher are the differences between SinG and MUNICH street concentrations respectively.

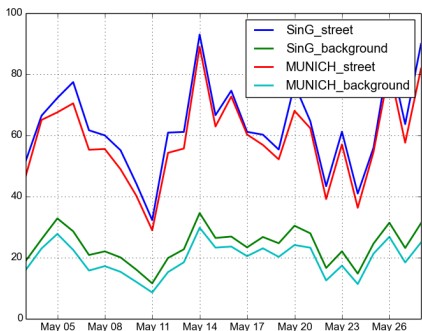

**Figure 15.** $NO_2$ daily concentrations in the street and in the background at CELES traffic station.

However, as indicated in Table 7, the magnitude of the differences between SinG and MUNICH depends very much on
the street: the lowest differences between SinG and MUNICH $NO_2$ concentrations are simulated at the stations BONAP and BP_EST, with differences below 3%, while the highest differences are simulated at the stations CELES, SOULT and ELYS, with differences around 10%.

To understand why the dynamic coupling between the background and the streets differs depending on stations, the differences between SinG and MUNICH are analysed in terms of the daily weighted mass fluxes that influence the street con-
centrations. As detailed in section 2.2, the street concentrations are influenced by the vertical mass flux from/to background concentrations ($Q_{vert}$), but also the emission mass flux ($Q_{emis}$) and the mass fluxes from the street lateral boundaries ($Q_{inflow}$, $Q_{outflow}$). Daily weighted mass fluxes ($qf_i$) are calculated according to:

$$q_{f_i} = \frac{Q_i}{\sum Q_i}; \quad \text{with} \quad \sum Q_i = Q_{inflow} + Q_{emis} + Q_{outflow} + Q_{vert} \quad (12)$$



Figure 16 shows the daily mass fluxes influencing the street concentrations at BONAP, CELES and BP_EST. At BONAP,
advection (inlet and outlet fluxes in Figure 16) dominates over vertical transfer, probably because the value of $\alpha_r$ is high, indi-
cating that the street is narrow. At BP_EST, Figure 16 indicates that vertical transfer is the dominant process. This dominance
of vertical transfer is because the street is large and the value of $\alpha_r$ is low. Note that BP_EST station also presents a high
emission flux, common data to both models SinG and MUNICH. Also, both BP_EST and BONAP present a low number of
connected streets, which may indicate an inferior vertical mass flux intersections compared to other traffic stations. At CELES,
where the value of $\alpha_r$ is intermediate, the inlet, outlet and vertical fluxes have the same order of magnitude, and the differences
between MUNICH and SinG are larger than at BONAP and BP_EST stations.

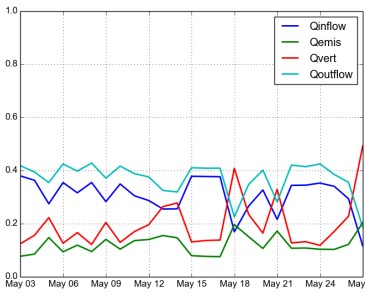 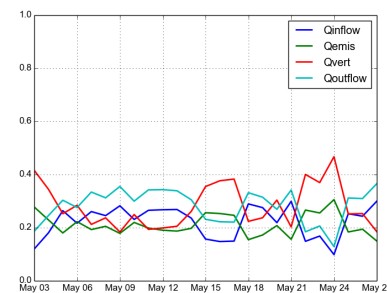 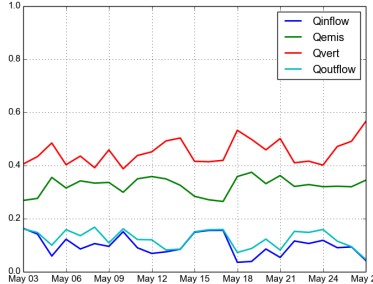

**Figure 16.** Daily weighted mass fluxes of $NO_2$ at BONAP (left panel), CELES (middle panel) and BP_EST (right panel) traffic stations.

NO concentrations are less sensitive to the dynamic coupling between local and regional scales than $NO_2$ concentrations,
and the average concentrations simulated with SinG and MUNICH are very similar at all stations (as indicated in Annexe A2).
This is explained by three reasons: $(i)$ NO background concentrations are very low compared to NO concentrations in streets;
$(ii)$ NO has a short lifetime, as it quickly reacts to form $NO_2$; and $(iii)$ NO concentrations in streets are mainly determined
by direct emissions, which are the same in MUNICH and SinG simulations. Figure 17 shows the daily-weighted mass fluxes
influencing the street concentrations at BONAP, CELES and BP_EST. At all three stations, the emission mass flux clearly
dominates over the inlet/outlet and vertical mass fluxes, confirming the strong and local influence of NO emissions on NO
concentrations.

To summarize, for NO concentrations, the dynamic coupling between the regional and local scales tends not to be important.
However, for $NO_2$ concentrations, it seems to be more important at stations with low to intermediate values of $\alpha_r$, where
the inlet, outlet and vertical fluxes have the same order of magnitude. In opposition, the dynamic coupling seems to be less
important at stations with low or high values of $\alpha_r$, where either the vertical flux or the inlet/oulet fluxe dominates the other.

To better quantify the importance of the dynamic coupling on the street concencentrations, the concentrations simulated
with SinG and MUNICH in each street are compared over the whole Paris city street network. The relative differences be-
tween concentrations simulated with the two models are computed in each street. The average over all streets of these relative
differences, as well as the minimum and maximum values are estimated and discussed below.

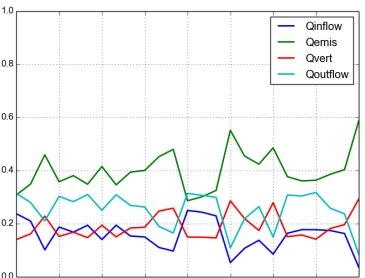 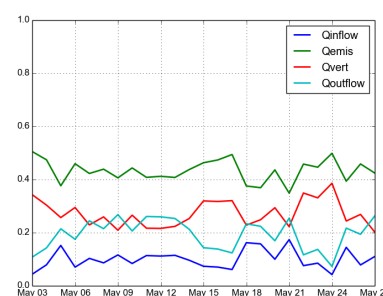 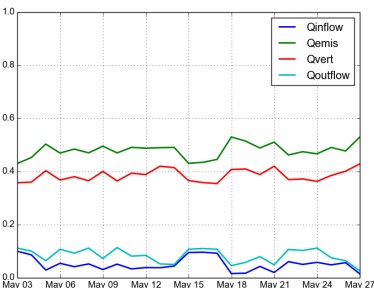

**Figure 17.** Daily weighted mass flux of NO at BONAP (left panel), CELES (middle panel) and BP_EST (right panel) traffic stations.

NO, NO$_2$ and NO$_x$ average concentrations simulated with SinG, as well as the mean relative differences between SinG and MUNICH are represented in Annexe B, in Figure B1. As it was observed at traffic stations, the average NO$_2$ concentrations are larger with SinG than MUNICH for most streets in the network, with an average relative difference over all streets of about 7.5%. Although this relative difference is low, the maximum and the minimum differences are high and reach 63% and $-28\%$ respectively. The average NO concentrations is slightly lower with SinG than MUNICH, the average relative difference over all streets is low and about -0.85%. As for NO$_2$, for NO concentrations, there is a large variation between the maximum and minimum differences (58% and $-35\%$ respectively). Particularly, NO concentrations simulated with SinG are generally lower than those simulated with MUNICH in the center of the street network. However, in other places, such as the ring road, NO concentrations simulated with SinG are about 5% higher than those simulated with MUNICH. Similarly to NO$_2$, NO$_x$ concentrations also presented low average differences between SinG and MUNICH, about 5% in the whole street-network, but with high maximum and minimun values (60% and $-27\%$ respectively). As discussed at the beginning of this section, relative differences between NO$_2$, NO and NO$_x$ concentrations simulated with SinG and MUNICH are strongly correlated to the emissions in the street and to the street aspect ratio $\alpha_r$. Therefore, large differences between SinG and MUNICH are observed in streets with high traffic emissions and intermediate to low values of $\alpha_r$, such as in the ring road, where the vertical mass transfer between streets and the background is important. The differences are less pronounced for NO concentrations, because of the short lifetime of NO.

As the majority of parisian streets presents an intermediate value of the street aspect ratio $\alpha_r$, to better understand the influence of the street aspect ratio on the dynamic coupling, the variations of the relative differences between NO$_2$ and NO concentrations simulated with SinG and MUNICH with the street aspect ratio $\alpha_r$ are studied. For the different ranges of $\alpha_r$ encountered in the street network, and for different ranges of relative differences, Figure 18 represents the percentage of streets involved in the network. Thus, in the figure, the sum of each column is 100%. In accordance with Figure 14, NO$_2$ average concentrations are in general higher using SinG than using MUNICH. The relative difference is mostly between 2% and 30% for streets with $\alpha_r$ smaller than 1.8, and between 2% and 10% for streets with $\alpha_r$ larger than 1.8. The higher the value of $\alpha_r$ is, the lower is the variability of relative differences. However, even for $\alpha_r$ larger than 1.8, relative differences between 10%





and 20% are relatively frequent (between 16% and 20% of the streets), indicating the influence of other factors than the street aspect ratios.

For NO, the average concentrations simulated with SinG are in general smaller than those simulated with MUNICH, mostly
between 0% and -10%. As for $NO_2$, the variability of relative differences is higher for low to intermediate values of $\alpha_r$.

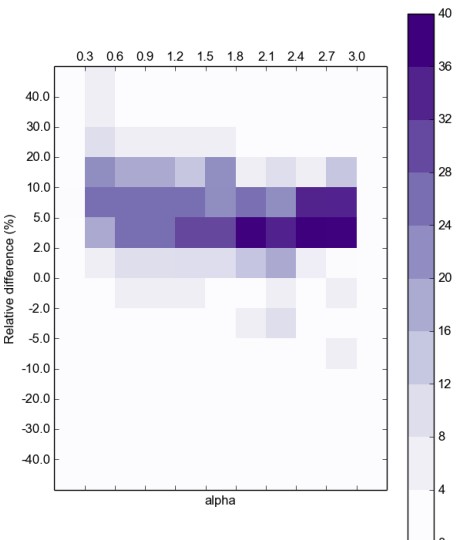

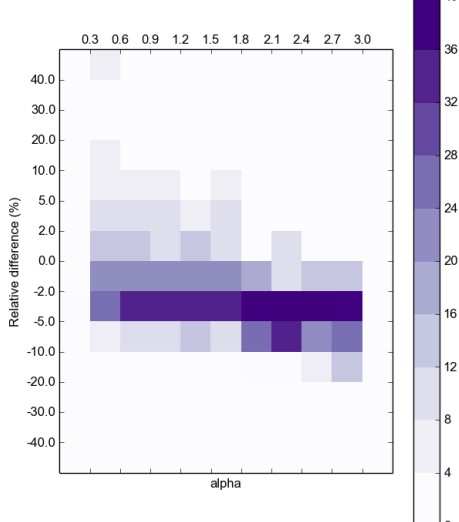

**Figure 18.** Percentage of streets present in each $\alpha_r$ interval according to $\alpha_r$ values and the $NO_2$ (left panel) and NO (right panel) relative differences between pollutant concentrations calculated by SinG and MUNICH.

## 5.2 Regional scale

Figure B2 represents the spatial distribution of average background $NO_2$ and $NO_x$ concentrations simulated with SinG, and the relative differences to those simulated with Polair3D. As indicated in section 4.2, background concentrations at the regional scale are influenced by the dynamic coupling with the local scale. $NO_2$ concentration differences between SinG and Polair3D
are in average 11%, with a maximum value equals to 34%. For $NO_x$ concentrations, the relative differences are of the same order of magnitude than for $NO_2$, with an average and a maximum value equal to 15% and 42% respectively. NO concentrations are not shown in Figure B2, because they are very low at the regional scale.

For both $NO_2$ and $NO_x$, the most important differences between Polair3D and SinG background concentrations are observed at the ring road and in the north-west of Paris city. Similarly to the local scale, relative differences of concentrations simulated
with SinG and MUNICH are higher in regions with high traffic emissions and where streets present an intermediate value of





$\alpha_r$, such as ELYS (see Figure 6). Note that, as mentioned in section 2.3, SinG output concentrations at the regional scale are an average of background and street concentrations in each grid cell. This justifies the higher differences between coupled and non-coupled simulations at the regional scale than at the local scale.

## 6   Conclusions

In this study, a Street-in-Grid (SinG) multi-scale simulation is performed over Paris city, with a dynamic coupling between the local (street) and regional (background) scales. For Paris, 3819 streets are considered and different databases are used to determine the width and height of each street. A stationary approach may be used to compute pollutant concentrations in the streets, by performing a mass balance between emission, deposition and vertical and horizontal transfer. Although this approach is reasonable to estimate $NO_x$ concentrations or the concentration of inert pollutants, it is not appropriate to compute

the concentrations of reactive pollutants such as $NO_2$ or NO. A non-stationary dynamic approach coupling finely chemistry and transport of pollutants was implemented and proved to be numerically stable. It leads to $NO_2$ and $NO_x$ concentrations that compare well to observations, both at the regional and local scales.

In the streets, $NO_x$ and $NO_2$ concentrations simulated by SinG compare well to measurements performed at traffic stations. For $NO_2$ concentrations, the statistical indicators obtained with SinG and the street model (MUNICH) respect the most strict

performance criteria (Hanna and Chang, 2012) at traffic stations. However, NO concentrations are strongly underestimated at traffic stations located in streets that converge in big squares. This underestimation is probably due to the short life time of NO, for which the assumption of uniform concentrations in wide streets and big squares may not be verified. At the regional scale, SinG performs also well for simulating $NO_x$ and $NO_2$ concentrations, and the most strict critera are respected at background stations.

The dynamic coupling between the regional and local scales is assessed by comparing the concentrations simulated with SinG to those simulated with MUNICH. $NO_x$ and $NO_2$ concentrations simulated with SinG and MUNICH are strongly correlated to traffic emissions, and the highest concentrations are observed in the ring road around Paris city ("boulevard périphérique"), where emissions are the highest. Similarly, at both the local and regional scales, the influence of the dynamic coupling is larger in areas where traffic emissions are high. $NO_2$ concentrations simulated with SinG are in general larger

than those simulated with MUNICH, especially in high emission areas, because the background concentrations in SinG are influenced by the high $NO_x$ concentrations of the street network. The influence of the dynamic coupling depends not only on the emission strenght, but also on the aspect ratio (height over width) of the street. Although, on average over the streets of Paris, the influence of the dynamic coupling on $NO_2$ concentrations in the street is only 7.5%, it can reach values as high as 63%. The influence of the dynamic coupling on background regional $NO_2$ concentrations can be large as well: 11% on average

over Paris with a maximum relative difference of 34%.

Further work will include the development of a new version of SinG to estimate particle-phase concentrations, taking into account the formation of secondary aerosols.





## Appendix A: Statistical parameters

### A1  Definitions

- FB: Fractional bias

$$FB = 2\left(\frac{\bar{o}-\bar{c}}{\bar{o}+\bar{c}}\right)$$

- MG: Geometric mean bias

$$MG = exp(\overline{ln(o)} - \overline{ln(c)})$$

- NMSE: Normalized mean square error

$$NMSE = \frac{\overline{(o-c)^2}}{\overline{oc}}$$

- VG: Geometric variance

$$VG = exp[\overline{(ln(o)-ln(c))^2}]$$

- NAD: Normalised absolute difference

$$NAD = \frac{\overline{|c-o|}}{(\bar{c}+\bar{o})}$$

- FAC2: Fraction of data that satisfy

$$0.5 \le \frac{c}{o} \le 2.0$$

- Correlation

$$cor = \frac{\overline{(o-\bar{o})(c-\bar{c})}}{\sigma_c \sigma_o}$$

Where $o$ and $c$ represent the observed and simulated concentrations respectively.



**A2   Statistical parameters at all traffic stations**

| | | NO2 | | | | | | | | NO | | | | | | | | NO$_x$ | | | | | | |
|---|---|---|---|---|---|---|---|---|---|---|---|---|---|---|---|---|---|---|---|---|---|---|---|---|---|
| | | o | s | FB | MG | NMSE | VG | FAC2 | NAD | o | s | FB | MG | NMSE | VG | FAC2 | NAD | o | s | FB | MG | NMSE | VG | FAC2 | NAD |
| CELES | Polair3D | 55.8 | 19.5 | -0.96 | 0.36 | 1.41 | 3.02 | 0.04 | 0.48 | 49.6 | 3.0 | -1.77 | 0.06 | 18.97 | 1590.06 | 0.00 | 0.88 | 131.5 | 24.1 | -1.38 | 0.19 | 4.50 | 15.44 | 0.04 | 0.69 |
| | MUNICH | 55.8 | 59.3 | 0.06 | 1.10 | 0.06 | 1.10 | 0.96 | 0.10 | 49.6 | 52.0 | 0.05 | 1.18 | 0.19 | 1.35 | 0.80 | 0.18 | 131.5 | 139.0 | 0.05 | 1.14 | 0.12 | 1.20 | 0.96 | 0.14 |
| | SinG | 55.8 | 64.0 | 0.13 | 1.19 | 0.08 | 1.13 | 0.96 | 0.12 | 49.6 | 51.6 | 0.04 | 1.17 | 0.21 | 1.37 | 0.80 | 0.19 | 131.5 | 143.1 | 0.08 | 1.18 | 0.13 | 1.23 | 0.88 | 0.15 |
| BONAP | Polair3D | 46.2 | 21.0 | -0.75 | 0.45 | 0.72 | 1.98 | 0.20 | 0.37 | 43.7 | 3.4 | -1.71 | 0.07 | 11.76 | 818.11 | 0.00 | 0.85 | 113.1 | 26.2 | -1.24 | 0.23 | 2.71 | 9.41 | 0.00 | 0.62 |
| | MUNICH | 46.2 | 53.6 | 0.15 | 1.15 | 0.07 | 1.07 | 1.00 | 0.11 | 43.7 | 25.9 | -0.51 | 0.58 | 0.37 | 1.47 | 0.68 | 0.25 | 113.1 | 93.4 | -0.19 | 0.81 | 0.09 | 1.10 | 1.00 | 0.12 |
| | SinG | 46.2 | 54.3 | 0.16 | 1.17 | 0.07 | 1.07 | 1.00 | 0.11 | 43.7 | 25.0 | -0.54 | 0.56 | 0.41 | 1.52 | 0.68 | 0.27 | 113.1 | 92.7 | -0.20 | 0.81 | 0.09 | 1.10 | 1.00 | 0.12 |
| SOULT | Polair3D | 40.4 | 20.7 | -0.64 | 0.51 | 0.55 | 1.63 | 0.48 | 0.32 | 19.6 | 3.3 | -1.41 | 0.19 | 5.52 | 18.29 | 0.00 | 0.70 | 70.3 | 25.8 | -0.92 | 0.38 | 1.33 | 2.72 | 0.12 | 0.46 |
| | MUNICH | 40.4 | 42.8 | 0.06 | 1.05 | 0.07 | 1.07 | 1.00 | 0.10 | 19.6 | 20.5 | 0.04 | 1.13 | 0.18 | 1.19 | 0.92 | 0.17 | 70.3 | 74.3 | 0.05 | 1.08 | 0.09 | 1.09 | 1.00 | 0.12 |
| | SinG | 40.4 | 46.1 | 0.13 | 1.14 | 0.08 | 1.08 | 1.00 | 0.11 | 19.6 | 20.1 | 0.02 | 1.12 | 0.16 | 1.17 | 0.92 | 0.16 | 70.3 | 77.0 | 0.09 | 1.12 | 0.08 | 1.09 | 1.00 | 0.12 |
| ELYS | Polair3D | 51.0 | 23.3 | -0.74 | 0.45 | 0.74 | 2.02 | 0.32 | 0.37 | 38.4 | 4.1 | -1.61 | 0.11 | 9.01 | 156.53 | 0.00 | 0.80 | 109.8 | 29.6 | -1.15 | 0.27 | 2.31 | 6.27 | 0.12 | 0.57 |
| | MUNICH | 51.0 | 45.5 | -0.11 | 0.89 | 0.07 | 1.08 | 1.00 | 0.12 | 38.4 | 19.4 | -0.66 | 0.53 | 0.76 | 1.80 | 0.56 | 0.35 | 109.8 | 75.2 | -0.37 | 0.70 | 0.26 | 1.27 | 0.84 | 0.22 |
| | SinG | 51.0 | 49.8 | -0.02 | 0.97 | 0.05 | 1.05 | 1.00 | 0.09 | 38.4 | 18.5 | -0.70 | 0.51 | 0.83 | 1.86 | 0.40 | 0.36 | 109.8 | 78.1 | -0.33 | 0.73 | 0.22 | 1.27 | 0.84 | 0.20 |
| OPERA | Polair3D | 74.3 | 23.6 | -1.03 | 0.31 | 1.55 | 4.00 | 0.00 | 0.51 | 81.1 | 4.1 | -1.80 | 0.05 | 19.20 | 7472.94 | 0.00 | 0.90 | 198.5 | 30.0 | -1.47 | 0.15 | 5.11 | 38.59 | 0.00 | 0.73 |
| | MUNICH | 74.3 | 56.7 | -0.26 | 0.75 | 0.11 | 1.13 | 1.00 | 0.14 | 81.1 | 29.5 | -0.93 | 0.36 | 1.27 | 3.04 | 0.16 | 0.46 | 198.5 | 102.1 | -0.64 | 0.51 | 0.54 | 1.67 | 0.48 | 0.32 |
| | SinG | 74.3 | 60.3 | -0.20 | 0.80 | 0.08 | 1.09 | 1.00 | 0.12 | 81.1 | 27.7 | -0.98 | 0.34 | 1.43 | 3.41 | 0.08 | 0.49 | 198.5 | 102.8 | -0.63 | 0.51 | 0.52 | 1.64 | 0.52 | 0.31 |
| HAUS | Polair3D | 56.1 | 23.3 | -0.82 | 0.42 | 0.98 | 2.25 | 0.28 | 0.41 | 37.2 | 4.0 | -1.60 | 0.12 | 10.00 | 109.89 | 0.00 | 0.80 | 112.8 | 29.5 | -1.16 | 0.27 | 2.67 | 6.08 | 0.08 | 0.58 |
| | MUNICH | 56.1 | 51.8 | -0.08 | 0.94 | 0.10 | 1.07 | 1.00 | 0.12 | 37.2 | 21.2 | -0.54 | 0.64 | 0.81 | 1.62 | 0.68 | 0.31 | 112.8 | 84.4 | -0.28 | 0.78 | 0.29 | 1.22 | 0.88 | 0.20 |
| | SinG | 56.1 | 55.5 | -0.01 | 1.00 | 0.09 | 1.07 | 1.00 | 0.11 | 37.2 | 19.8 | -0.60 | 0.60 | 0.92 | 1.71 | 0.60 | 0.33 | 112.8 | 86.0 | -0.27 | 0.80 | 0.28 | 1.21 | 0.88 | 0.20 |
| BP_EST | Polair3D | 70.7 | 24.2 | -0.97 | 0.37 | 1.79 | 3.40 | 0.32 | 0.49 | 88.6 | 4.5 | -1.80 | 0.06 | 26.11 | 2997.77 | 0.00 | 0.90 | 206.3 | 31.2 | -1.47 | 0.18 | 6.89 | 29.36 | 0.12 | 0.73 |
| | MUNICH | 70.7 | 81.7 | 0.14 | 1.26 | 0.20 | 1.38 | 0.80 | 0.18 | 88.6 | 84.5 | -0.04 | 1.27 | 0.43 | 2.29 | 0.64 | 0.26 | 206.3 | 211.4 | 0.02 | 1.24 | 0.31 | 1.77 | 0.64 | 0.22 |
| | SinG | 70.7 | 80.3 | 0.12 | 1.24 | 0.20 | 1.38 | 0.80 | 0.18 | 88.6 | 81.5 | -0.08 | 1.22 | 0.45 | 2.27 | 0.56 | 0.27 | 206.3 | 205.2 | -0.005 | 1.21 | 0.32 | 1.76 | 0.64 | 0.23 |
| BASCH | Polair3D | 78.4 | 20.0 | -1.18 | 0.25 | 2.37 | 7.42 | 0.00 | 0.59 | 98.1 | 3.1 | -1.86 | 0.03 | 30.1 | 115444.50 | 0.00 | 0.93 | 228.9 | 25.0 | -1.60 | 0.11 | 7.82 | 157.58 | 0.00 | 0.80 |
| | MUNICH | 78.4 | 50.0 | -0.44 | 0.63 | 0.28 | 1.33 | 0.80 | 0.22 | 98.1 | 26.8 | -1.14 | 0.27 | 2.16 | 5.79 | 0.00 | 0.57 | 228.9 | 91.1 | -0.86 | 0.39 | 1.04 | 2.55 | 0.20 | 0.43 |
| | SinG | 78.4 | 51.5 | -0.41 | 0.65 | 0.25 | 1.30 | 0.80 | 0.20 | 98.1 | 25.7 | -1.16 | 0.26 | 2.32 | 6.39 | 0.00 | 0.58 | 228.9 | 90.9 | -0.86 | 0.39 | 1.04 | 2.55 | 0.16 | 0.43 |





## Appendix B: Concentration maps - local and regional scales

### B1    Local scale



**Figure B1.** NO$_2$ (top panels), NO (middle panels) and NO$_x$ (bottom panels) concentrations simulated over Paris with SinG (left panels) and relative differences between SinG and MUNICH (right panels).

### B2    Regional scale





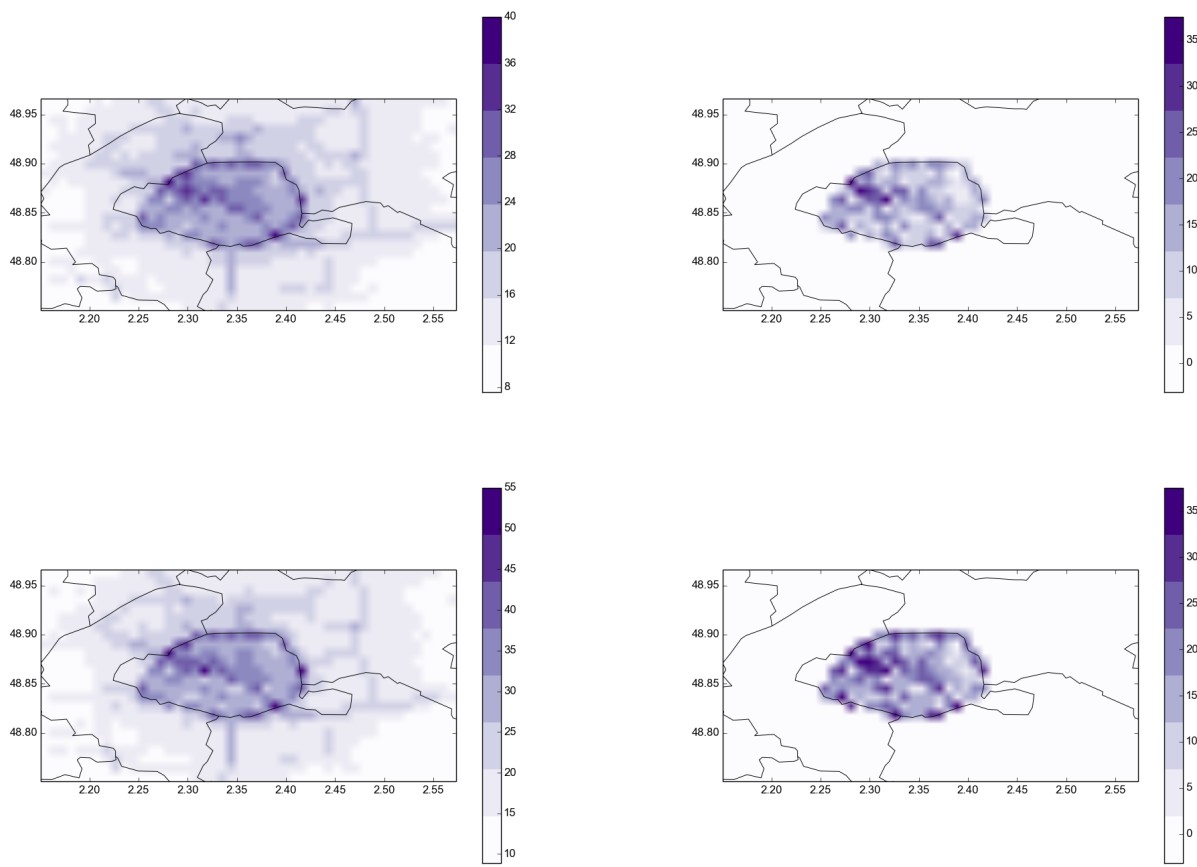

**Figure B2.** NO$_2$ (top panels) and NO$_x$ (bottom panels) concentrations simulated over Paris with SinG (left panels) and relative differences between SinG and Polair3D, in % (right panels).

*Acknowledgements.* This study was partially funded by the Departement of green spaces and environment of Paris City. The authors also thank Aiparif and the ANSES (french agency for food safety, environment and labor) working group on ambiant particulate matter for the trafic emission informations. Dr Yelva Roustan, Fabrice Dugay and Olivier Sanchez are gratefully acknowledged for discussions.





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
