# Peer review of "Street-in-Grid modeling of gas-phase pollutants in Paris city"

_Atmospheric Chemistry and Physics, 2019_

## Referee Comment (RC1) · Anonymous Referee #1 · 8 Jan 2020

1. General comments The objective of the paper is to quantify the effect of a dynamic multi-scale modeling between the regional and local scales on NO, NO2 and NOx concentrations over the street network of Paris city. This is done using a recently developed multi-scale model system named Street-in-Grid (SinG) that estimates gaseous pollutant concentrations simultaneously at local and regional scales, coupling them dynamically thereby addressing the question of double counting of emissions. This coupling combines the regional-scale chemistry-transport model Polair3D and the street network model MUNICH (Model of Urban Network of Intersecting Canyons and Highway). A new non-stationary approach is implemented for pollutant dispersion in streets with a fine coupling between transport and chemistry to improve prediction of the reactive pollutants of NO2 or NO.

[Figure]

The analysis covers a number of aspects (a) stationary versus non-stationary approach for different time steps, (b) model validation by comparing simulated and observed concentrations at both traffic and urban background stations of Paris city (c) the influence of the dynamic coupling between the regional and local scales.

The paper demonstrates improvements in model predictions when using the non-stationary approach and dynamic coupling approach based on model validation as well as analysis of model elements and inputs. Both approaches are novel compared to existing multi-scale model systems, and the paper provides a substantial contribution to scientific progress.

The paper is based on solid scientific methods.

The paper is very detailed in the analysis and subsequently relatively long.

The presentation is clear and the paper is well written and well structured. The conclusion is supported by the data presented, analysis and discussion.

2. Specific comments The authors should justify why only a relatively short period (1-28 May, 2014) is used for model validation. There is a mismatch between the year of emissions over Île-de-France of the domain 3 and over the domain 4 that is from 2012 and the model validation period of 2014. Explain how this may influence comparison of model results and measurements. Remove line 481-482 in the conclusion as the conclusion should not state future research endeavours.

3. Technical corrections Line 101 "DEOM" should be "DEHM" and "Operational" should be "Hemispheric". Consider to use a finer colour scale with more categories in Figure 4. Line 265 "The most important emissions" should be "The highest emissions". Figure 6, stations names should be larger to ease reading. To ease the reading of Table 6 two columns could be added that indicate which traffic stations have high traffic emissions and which are adjacent to big squares.
* * *
[Figure]

2019.

---

## Referee Comment (RC2) · Anonymous Referee #2 · 8 Jan 2020

General comments:

This manuscript presents recent developments of the multi-scale modelling system Street-in-Grid (SinG) which dynamically couples the mesoscale chemistry transport model Polair3D and the street network model MUNICH with two-way feedback. A new non-stationary numerical scheme is implemented in MUNICH that avoids the time step dependency in the partitioning of NO and NO2 chemistry. The new approach is used to evaluate SinG during May 2014 over Paris city and discuss the benefit of the two-way coupling between MUNICH and Polair3D when modelling NOx, NO and NO2. The SinG model adopts an elegant solution to avoid the double-counting of traffic emissions and is one of the few street-scale models that solve complex gas-phase chemistry based on Carbon Bond 2005 chemical mechanism. As stated by the Authors at the

end of the Conclusions, it will be extended in the near future to solve condensed phase chemistry. All these characteristics make SinG an excellent modelling tool to advance research on urban chemistry.

I have some general comments. The first one is about the title, which, in my opinion, is too generic and does not reflect the content of the manuscript. I suggest the Authors consider a reformulation of the title that better describes the main objective of the work. The main focus is on NOx/NO/NO2 representation, and the two-way feedback addressed in SinG. Regarding the new numerical scheme implemented in MUNICH and SinG, the discussion would benefit with some quantification of the computational time used in a stable stationary configuration compared with the new non-stationary solution presented in the manuscript. Is there any overhead added with the non-stationary approach? Are other gases apart from NOx sensitive to the old numerical scheme that makes the solution unstable or with a small enough time step the stationary solution is still accurate? The two-way feedback implemented in SinG is very elegant to avoid the double-counting of emissions at the urban scale, but it is somehow counter-intuitive the results compared with MUNICH alone which indeed has double-counting emissions from Polair3D background. One would expect MUNICH results to be overestimated due to the double-counting effect, but this is not the case. Both SinG and MUNICH evaluations with measurements are very similar. Some elaboration on the possible reasons for this result and the implications for other modelling systems that may still have double-counting of emissions in their urban solutions would be desired. Finally, some discussion about the impact of the Street-in-Grid at the regional scale downwind the city is missing. It is clear that the two-way coupling will improve the skills of SinG at the regional scale if it is evaluated with urban sites, but does this result also in an improvement of the mesoscale model photochemistry downwind Paris? Is there any sensitivity in NOx and other reactive gases like O3 in some rural areas affected by the pollution plume of Paris?

The results of the manuscript are novel and have an interest in the scientific community. However, I have the impression that the material presented is more suited for the "Geoscientific Model Development" than "Atmospheric Chemistry and Physics" journal. Overall, the manuscript is well written but deserves some English editing. I recommend the authors to address the general comments and improve the manuscript following the specific and technical comments detailed below.

Specific comments:

- Line 1: Quantify or provide a range for "coarse spatial resolution".

- Line 15: I suggest to explicitly mention in the abstract that SinG implements a two-way feedback. The Authors could use "two-way dynamical coupling" or "a dynamical coupling between the regional and local scales with a two-way feedback."

- Line 74: The concept of dynamic coupling defined here is confusing. In multi-scale or nested domain models, the dynamic coupling can be one-way or two-way. The latter means that the feedback from the smaller scale to the coarser scale is allowed, which is the case of SinG. For the sake of clarity, I recommend using the concept of "two-way dynamic coupling" or "dynamic coupling with two-way feedback".

- Line 108: The sentence explaining how the multi-scale concentrations are obtained in Stocker et al. (2012) is not clear. What is the difference between the "gridded concentration" and the "regional-scale concentration"?

- Line 113: The objective is very well presented here; part of this sentence could be used to improve the current manuscript Title. I think that the novel contribution of the work is the discussion on the role of the two-way feedback between scales.

- Line 124: Is SinG a model or an interface? The Authors could clarify how MUNICH and Polair3D are integrated into SinG. Is SinG a version of Polair3D with an urban component that runs MUNICH internally as a subroutine providing meteorological and chemistry inputs?

- Line 128: A one-way formulation is still a dynamic coupling. As suggested before, the

use of "two-way feedback" may help the reader understand the added value of SinG compared with other modelling systems.

- Line 147: What are the implications of assuming the concentrations uniform within the street segments? What is the maximum length of a street segment allowed in MUNICH?

- Line 172: How is the standard deviation of the vertical wind speed computed from WRF variables?

- Line 174: The CB05 is a gas-phase mechanism. Please, replace "concentration of pollutants" for "concentration of gases" and "module" for "mechanism".

- Line 184: In equation 9, is the parameter triangle sub-zero the same as triangle sub-one but for time n? Please, clarify the meaning of the notation used to define triangle sub one.

- Line 193: What is the computational overhead of running MUNICH coupled to Polair3D in SinG compared with running only Polair3D?

- Line 195: Clarify in the text that "cell i" is the cell of the regional model.

- Line 203: From equation 11, SinG does not perform an average but a sum of the mass of background and the street. Please, amend. Why the authors use Vcell instead of (Vcell-Vbuild)? This is the exact volume from where equation 11 derives the mass.

- Line 206: I guess there is an error in the numbers of sub-Sections 2.4, 2.5, 2.6 and 2.7. sub-Section 2.4 should be a new Section 3, and the following sub-sections the new 3.1, 3.2 and 3.3.

- Line 208: Did the authors perform any spin-up in the chemistry?

- Line 210: Please clarify if SinG runs the 4 Polair3D domains or it is a decoupled run? Is Polair3D using two-way nesting?

- Line 240: Not many streets are used in the local-scale domain. What are the implications in the total traffic emissions of Paris ingested in the models then? Can the Authors quantify the percentage of emissions that the main streets used in the simulations represent from the total? How are the rest of the streets treated SinG during the two-way coupling? Are all the streets still used in eq 10 for Vbuild_i or only the main streets?

- Line 268: What is the temporal resolution of the background concentrations used in MUNICH? This may explain part of the differences seen between SinG and MUNICH. SinG may use background conditions with higher temporal variability compared with MUNICH set up.

- Line 292: Why the authors consider that some of the results are not stable numerically? None of the runs shows numerical instabilities, at least from what can be seen in Figure 7 and 8. The solution using 600s is quite similar to the one with 100s. Which are the criteria to identify numerical instabilities in SinG or MUNICH?

- Line 293: Please, show the observations in Figure 7 and 8. Why do the Authors not show SinG and MUNICH results in the same station both figures? It is difficult to appreciate the differences between methodologies using different sites.

- Line 295: Why a time step of 100s is selected with the non-stationary approach? Figure 7 and 8 show the same results with 600s, which imply that the model should be much faster with the same accuracy with 600s.

- Line 304: Please, provide MUNICH results in Table 5 or clarify if MUNICH results are the same as the background concentration of Polair3D in open-areas.

- Line 325: Figure 11 shows a better agreement of SinG and MUNICH during the morning peak than the evening one. Do the Authors have an explanation for this behaviour?

- Line 330: The Polair3D shows a substantial underestimation of NO as most CTMs. It appears a significant drawback for MUNICH and SinG to reproduce NO in open areas

within the city. Can the Authors elaborate on approaches to overcome this limitation?

- Line 441: Results of SinG at the regional scale over the Paris area show higher NOx concentrations than Polair3D. What is the impact downwind Paris area in some urban background or rural sites using SinG or Polair3D? Does this increase of NOx in SinG results in a positive effect on the model downwind Paris (i.e., O3)? If this is the case, it would be relevant to elaborate on this because it has implications in the way how urban-cities are modeled in mesoscale models.

- Line 452: As mentioned before, equation 11 does not define SinG regional scale concentration as an average but a sum of masses.

- Line 480: I suggest to add a last sentence highlighting that NO is less sensitive to this coupling and why.

- Line 481: Do the Authors have any plan to evaluate other gases at street-level in the future? One of the most important capabilities of SinG is solving complex chemistry at street-scale. Understand the dynamics of other reactive gases in the urban environment deserves future research efforts.

Technical comments:

- Figure and Table captions: all captions should be self-explanatory. Several Tables and Figures present information that is not described in the caption (i.e., name of variables, units, the meaning of acronyms or abbreviations.)

- Equations: There are several equations with the definition of terms in the same line. Please, split those cases in separate equations. This occurs in Eq 3, 7, 8, 9, 11.

- Line 2: Delete "i.e."

- Line 15: Amend the format of the World Health Organization reference. Use "WHO (2016)" or "World Health Organization (2016)". It has not much sense to define an acronym that will not be used anymore in the text.

- Line 25: Add a comma after "e.g." or "i.e.".

- Line 25: Follow the appropriate style used by ACP when using references between parentheses. To simplify the text, I suggest using only the references, not the model acronyms.

- Line 44: Use the same style to introduce acronyms throughout the text, first the complete name followed by the acronym in parentheses.

- Line 44: Avoid opening a parenthesis just after a closing one, and use similar style format if several references are provided. In particular, "(CALINE4; Bensons, 1984; Sharma, et al.). Check and unify the reference format used in the text.

- Line 44: The reference "Sharma et al." is incomplete. Please, provide the year here and in the reference section.

- Line 52: Define the acronym SIRANE, as done previously with other models.

- Line 136: Amend the use of references. In this case should be like "at different locations (e.g., Sartelet et al., 2012; Abdallah et al., 2018; ...)."

- Line 137: Remove "including Greater Paris" from the middle of the list of references.

- Line 150, Equation 2: The use of Q and Qinflow are confusing. Q is a mass variable while Qinflow, Qemis, Qoutflow, ..., Qdep are mass fluxes. I suggest using another letter for the fluxes, e.g., Finflow.

- Line 153: Amend "each of this term" with "each term".

- Line 158: Please, define what is Qair and Cst like is done with H, W and Ust.

- Line 161: Together with the reference of equation 8 from Kim et al. (2018), add the reference to equation 7 of this manuscript.

- Line 194: Amend "details" with "detailed".

- Line 220: Specify the version of WRF model used.

- Line 222: Is the top of the atmosphere in WRF set at 5000m? If not, amend.

- Line 225: Delete "and chemical".

- Line 228: Specify the version of MEGAN model used.

- Line 233: Amend the link provided. It is not working.

- Line 236: Specify the period of average used in Figure 4.

- Line 284: Figure 6 and Figure 5 could be combined in a single figure. I recommend using two different colours to differentiate urban and traffic measurement stations (i.e., red and black dots).

- Line 358: Please, clarify how relative difference is computed. Which is the reference value?

- Line 361: It would be desirable to use the same range and colour scale in both panels of Figure 14.

- Line 389: Amend "daily mass fluxes" with "daily weighted mass fluxes".

- Line 396: In the legends of Figure 16, use the same notation as equation 12. Instead of Qinflow should be qf_inflow.

- Line 401: Check the use of the hyphen. Sometimes is used and others not (e.g., daily weighted, daily-weighted).

- Line 439: Detailed the meaning of the colour in the Figure by adding "Percentage of streets (purple colour). Add a % at the top of the colour scale.

- Line 460: Re-word "coupling finely".

- Line 467: Use "appropriate" instead of "verified".

- Line 497: Correlation is never used in the manuscript. Please, remove the statistic from the Annex.

---

## Author Comment (AC1) · 17 Mar 2020

Anonymous Referee #1 General comments

The objective of the paper is to quantify the effect of a dynamic multi-scale modeling between the regional and local scales on NO, NO2 and NOx concentrations over the street network of Paris city. This is done using a recently developed multi-scale model system named Street-in-Grid (SinG) that estimates gaseous pollutant concentrations simultaneously at local and regional scales, coupling them dynamically thereby addressing the question of double counting of emissions. This coupling combines the regional-scale chemistry-transport model Polair3D and the street network model MUNICH (Model of Urban Network of Intersecting Canyons and Highway). A new

non-stationary approach is implemented for pollutant dispersion in streets with a fine coupling between transport and chemistry to improve prediction of the reactive pollutants of NO2 or NO. The analysis covers a number of aspects (a) stationary versus non-stationary approach for different time steps, (b) model validation by comparing simulated and observed concentrations at both traffic and urban background stations of Paris city (c) the influence of the dynamic coupling between the regional and local scales. The paper demonstrates improvements in model predictions when using the non-stationary approach and dynamic coupling approach based on model validation as well as analysis of model elements and inputs. Both approaches are novel compared to existing multi-scale model systems, and the paper provides a substantial contribution to scientific progress. The paper is based on solid scientific methods. The paper is very detailed in the analysis and subsequently relatively long. The presentation is clear and the paper is well written and well structured. The conclusion is supported by the data presented, analysis and discussion.

Specific commentsÂă:

- The authors should justify why only a relatively short period (1-28 May, 2014) is used for model validation.

Reply: This paper aims at analyzing the influence of the non-stationary regime and multi-scale coupling at both local and regional scales. Many runs were performed for this sensitivity study, and a one-month simulation period is long enough to analyze the processes.

- There is a mismatch between the year of emissions over Île-de-France of the domain 3 and over the domain 4 that is from 2012 and the model validation period of 2014. Explain how this may influence comparison of model results and measurements.

Reply: As specified in the paper, 2012 Airparif inventory is used only for sectors different than road traffic. Traffic emissions use data specific of 2013 and 2014. Comparisons between the 2012 Airparif inventory and the more recent 2015 Airparif inventory

show that the most important differences in NOx emissions between the two years are due to differences in traffic emissions. Because traffic emissions are specific of the year studied here, we do not think that using the 2012 inventory for sources other than traffic impact our comparison of model results to measurements.

- Remove line 481-482 in the conclusion as the conclusion should not state future research endeavours. Reply: These lines are removed.

Technical corrections

- Line 101 "DEOM" should be "DEHM" and "Operational" should be "Hemispheric". Done - Consider to use a finer colour scale with more categories in Figure 4. Done. - Line 265 "The most important emissions" should be "The highest emissions". Done - Figure 6, stations names should be larger to ease reading. Done. - To ease the reading of Table 6 two columns could be added that indicate which traffic stations have high traffic emissions and which are adjacent to big squares. Done.

---

## Author Comment (AC2) · 17 Mar 2020

General comments:

This manuscript presents recent developments of the multi-scale modelling system Street-in-Grid (SinG) which dynamically couples the mesoscale chemistry transport model Polair3D and the street network model MUNICH with two-way feedback. A new non-stationary numerical scheme is implemented in MUNICH that avoids the time step dependency in the partitioning of NO and NO2 chemistry. The new approach is used to evaluate SinG during May 2014 over Paris city and discuss the benefit of the twoway coupling between MUNICH and Polair3D when modelling NOx, NO and NO2. The

SinG model adopts an elegant solution to avoid the double-counting of traffic emissions and is one of the few street-scale models that solve complex gas-phase chemistry based on Carbon Bond 2005 chemical mechanism. As stated by the Authors at the end of the Conclusions, it will be extended in the near future to solve condensed phase chemistry. All these characteristics make SinG an excellent modelling tool to advance research on urban chemistry. I have some general comments. The first one is about the title, which, in my opinion, is too generic and does not reflect the content of the manuscript. I suggest the Authors consider a reformulation of the title that better describes the main objective of the work. The main focus is on NOx/NO/NO2 representation, and the two-way feedback addressed in SinG.

Reply: We modified the title from "Street-in-Grid modeling of gas-phase pollutants in Paris city" to "Non-stationary modeling of NO2, NO and NOx in Paris city using Street-in-Grid model: coupling local and regional scales with a two-way dynamic approach."

Regarding the new numerical scheme implemented in MUNICH and SinG, the discussion would benefit with some quantification of the computational time used in a stable stationary configuration compared with the new non-stationary solution presented in the manuscript. Is there any overhead added with the non-stationary approach?

Reply: The ratio of computational times observed using the non-stationary configuration compared to the stationary one was about 1.3 using both MUNICH and SinG models with the time-step 100s. This value was obtained with a machine with 256Go of RAM and processors Bi-Xeon E5-2650 v4 12 cores 2.2GHz.

Are other gases apart from NOx sensitive to the old numerical scheme that makes the solution unstable or with a small enough time step the stationary solution is still accurate?

Reply: Other species, such as O3 or VOCs, are sensitive to time step using the stationary approach. For example, regarding MUNICH results at the traffic station CELES, the O3 daily-average concentration during the whole simulated period increased by 8.5%

after changing the simulation time step from 600s to 100s. With the non-stationary approach, this difference passes to 0.07%. Furthermore, this O3 average concentration is higher with the non-stationary approach, passing from 52.1 $\mu$g/m3 using the stationary approach to 73.8 $\mu$g/m3 using the non-stationary approach (both values are obtained using a time-step of 100s). Similar differences were obtained with SinG. Some organic compounds are also sensitive to the time step with the stationary approach, but present similar concentrations with the stationary and non-stationary approaches with a time step of 100 s. For example, the daily-average concentration of Isoprene over the whole simulated period at CELES street passed from 1.26 $\mu$g/m3 to 1.39 $\mu$g/m3 after the time-step reduction using the stationary approach (increasing by 10.3%). Lastly, inert species as CO present the same results in stationary and non-stationary regimes, as mentioned in the manuscript.

The two-way feedback implemented in SinG is very elegant to avoid the double-counting of emissions at the urban scale, but it is somehow counter-intuitive the results compared with MUNICH alone which indeed has double-counting emissions from Polair3D background. One would expect MUNICH results to be overestimated due to the double-counting effect, but this is not the case. Both SinG and MUNICH evaluations with measurements are very similar. Some elaboration on the possible reasons for this result and the implications for other modelling systems that may still have double-counting of emissions in their urban solutions would be desired.

Reply: The double-counting of emissions in MUNICH simulations does not result in higher concentrations at the local scale compared to SinG. MUNICH simulations employ background concentrations calculated with Polair3D, i.e. no influence of the streets on the background, considering traffic emissions as surface emissions averaged over the grid cell. The two-way coupling performed by SinG allows mass transfer between local and regional scales, correcting background concentrations (as described in the section Street-in-Grid model). In streets where concentrations obtained by MUNICH and SinG are very similar, the vertical flux over the whole grid cell is close to the traffic

surface emissions. But this vertical flux can be larger than traffic surface emissions, for example when street concentrations are high due to important traffic emissions. In other words, streets with high traffic emissions tend to present high vertical mass flux, increasing background concentrations of SinG compared to Polair3D, and consequently street concentrations of SinG compared to MUNICH. This effect can be observed, for example, regarding the relative difference between NO concentrations in the streets (Figure B1). SinG concentrations are lower than those obtained by MUNICH in the center of Paris, but this relation changes in streets with very high emissions, as the boulevard périquérique (street-network ring road).

For clarity, the sentence "If the vertical mass transfer is high, then background concentrations may be higher in the two-way approach of SinG than in the one-way approach of MUNICH, leading to higher concentrations in streets." is added line 357 of the original version, after "In these areas, the vertical mass transfer between the local and regional scales tend to be more important for two main reasons: (i)... (ii) .... Intersections."

The influence of the two-way coupling is detailed in the paper and in the conclusion (lines 475-480), so that other modelling systems that may still have double-counting of emissions in their urban solutions may decide to develop or not a two-way coupling.

Finally, some discussion about the impact of the Street-in-Grid at the regional scale downwind the city is missing. It is clear that the two-way coupling will improve the skills of SinG at the regional scale if it is evaluated with urban sites, but does this result also in an improvement of the mesoscale model photochemistry downwind Paris? Is there any sensitivity in NOx and other reactive gases like O3 in some rural areas affected by the pollution plume of Paris?

Reply: O3 background concentrations obtained with SinG are in average 5.90% larger than those obtained by Polair3D, with a maximal value of 20%. These relative differences of O3 concentrations have a similar spatial distribution as observed in Figure B2

(right panel), limited mainly inside Paris city. No considerable differences are observed outside the street-network.

The results of the manuscript are novel and have an interest in the scientific community. However, I have the impression that the material presented is more suited for the "Geoscientific Model Development" than "Atmospheric Chemistry and Physics" journal. Overall, the manuscript is well written but deserves some English editing. I recommend the authors to address the general comments and improve the manuscript following the specific and technical comments detailed below.

Reply: The special issue to which this paper is submitted is a joint issue between "Geoscientific Model Development" and "Atmospheric Chemistry and Physics" journals. English was revised.

Specific comments:

- Line 1: Quantify or provide a range for "coarse spatial resolution".

Reply: Line 1 "Regional-scale chemistry-transport models have coarse spatial resolution, and thus can only simulate background concentrations." is replaced by "Regional-scale chemistry-transport models have coarse spatial resolution (coarser than 1 km x 1 km), and thus can only simulate background concentrations.

- Line 15: I suggest to explicitly mention in the abstract that SinG implements a two-way feedback. The Authors could use "two-way dynamical coupling" or "a dynamical coupling between the regional and local scales with a two-way feedback."

Reply: Line 6 of the original version: "This coupling combines the regional-scale chemistry-transport model Polair3D and the street network model MUNICH (Model of Urban Network of Intersecting Canyons and Highway)." is replaced by " This coupling combines the regional-scale chemistry-transport model Polair3D and the street network model Model of Urban Network of Intersecting Canyons and Highway (MUNICH) with a two-way feedback." Line 15 of the original version: "added value of multi-scale

modeling with a dynamical coupling between the regional and local scales." is replaced by: "added value of multi-scale modeling with a two-way dynamical coupling between the regional and local scales." Line 17 of the original version: "The dynamic coupling between the local and regional scales tends to be important for streets with an intermediate aspect ratio and with high traffic emissions." is replaced by: "The two-way dynamic coupling between the local and regional scales tends to be important for streets with an intermediate aspect ratio and with high traffic emissions."

- Line 74: The concept of dynamic coupling defined here is confusing. In multi-scale or nested domain models, the dynamic coupling can be one-way or two-way. The latter means that the feedback from the smaller scale to the coarser scale is allowed, which is the case of SinG. For the sake of clarity, I recommend using the concept of "two-way dynamic coupling" or "dynamic coupling with two-way feedback".

Reply: Line 74 of the original version: "Although MUNICH is able to consider the temporal and spatial evolution of background concentrations, the coupling between the background and street concentrations is not dynamic." is replaced by: "Although MU-NICH is able to consider the temporal and spatial evolution of background concentrations, the coupling between the background and street concentrations is not two-way, but one-way."

Line 76 of the original version "The coupling between background and street concentrations is dynamic in the multi-scale..." is replaced by "The coupling between background and street concentrations is two-way in the multi-scale...". Line 98 of the original version "For streets, several models consider a multi-scale modeling between streets and background concentrations, although this multi-scale is most often not dynamic." is replaced by: "For streets, several models consider a multi-scale modeling between streets and background concentrations, although this multi-scale is most often not two ways." Line 104 of the original version: "With this kind of non-dynamic multi-scale modeling, traffic emissions are counted twice:..." replaced by: show a fairly good agreement, especially for NO2 , whereas PM2.5 and PM10 are underestimated. With this

kind of one-way multi-scale modeling, traffic emissions are counted twice..." Line 113 of the original version: "The objective of this work is to quantify the effect of a dynamic multi-scale modeling between the regional and local scales on NO, NO2 and NOx concentrations over the street network of Paris city." is replaced by: "The objective of this work is to quantify the effect of a two-way dynamic multi-scale modeling between the regional and local scales on NO, NO2 and NOx concentrations over the street network of Paris city." Line 121 of the original version: "Finally, the fifth section studies the influence of the dynamic coupling between the regional and local scales." is replaced by: "Finally, the sixth section studies the influence of the two-way dynamic coupling between the regional and local scales."

- Line 108: The sentence explaining how the multi-scale concentrations are obtained in Stocker et al. (2012) is not clear. What is the difference between the "gridded concentration" and the "regional-scale concentration"?

Reply: Line 106 of the original version: "To avoid this double counting in multi-scale modeling, Stocker et al. (2012) used a different approach: the Gaussian model ADMS-Urban is applied to estimate the initial dispersion of traffic emissions during a mixing time $\tau$m (typically 1 hour). The multi-scale concentrations are obtained by subtracting the gridded concentrations simulated after this mixing time $\tau$m to the sum of the local-scale concentrations simulated with ADMS-Urban and the regional-scale concentrations." is replaced by: "To avoid this double counting in multi-scale modeling, Stocker et al. (2012) used a specific approach to couple the regional-scale model CMAQ and the local-scale Gaussian model ADMS-Urban. The local-scale effect of pollutant dispersion is calculated during a mixing time $\tau$m (typically 1h) by computing the differences in concentrations due to the dispersion of traffic emission using a gaussian and a non-gaussian approach on the spatial grid of CMAQ. Then the multi-scale concentrations are obtained by adding this local-scale effect to the CMAQ regional-scale concentrations."

- Line 113: The objective is very well presented here; part of this sentence could be

used to improve the current manuscript Title. I think that the novel contribution of the work is the discussion on the role of the two-way feedback between scales.

Reply: The original title "Street-in-Grid modeling of gas-phase pollutants in Paris city" is modified to "Non-stationary modeling of NO2, NO and NOx in Paris city using Street-in-Grid model: coupling local and regional scales with a two-way dynamic approach".

- Line 124: Is SinG a model or an interface? The Authors could clarify how MUNICH and Polair3D are integrated into SinG. Is SinG a version of Polair3D with an urban component that runs MUNICH internally as a subroutine providing meteorological and chemistry inputs?

Reply: Line 124 of the original version: "Street-in-Grid (SinG) is a multi-scale model that acts as an interface between the 3D chemistry-transport model Polair3D and the street-network model MUNICH (Model of Urban Network of Intersecting Canyons and Highways). MUNICH is coupled to the first vertical level of Polair3D and the mass transfer between the local and regional scales is computed at each time step. More details about the dynamic coupling are described in the section 3 of Kim et al. (2018) and in the section 2.3 of this paper." is replaced by: "Street-in-Grid (SinG) is a multi-scale model that couples the street-network Model of Urban Network of Intersecting Canyons and Highways (MUNICH) with the 3D chemistry-transport model Polair3D using a two-way dynamic multi-scale approach. MUNICH is coupled to the first vertical level of Polair3D and the mass transfer between the local and regional scales is computed at each time step of Polair3D. More details about the dynamic coupling are described in the section 3 of Kim et al. (2018) and in the section 2.3 of this paper."

- Line 128: A one-way formulation is still a dynamic coupling. As suggested before, the use of "two-way feedback" may help the reader understand the added value of SinG compared with other modelling systems.

Reply: Line 128 of the original version "This dynamic (two-way) coupling presents several advantages compared to a one-way formulation, as:" is replaced by: "This two-

way coupling presents several advantages compared to a one-way formulation, as:"

- Line 147: What are the implications of assuming the concentrations uniform within the street segments? What is the maximum length of a street segment allowed in MUNICH?

Reply: Assuming uniform concentrations within each street segments implies that street dimensions are constant in each segment. In each segment, because MUNICH is a stand-alone model, it does not have any constraint on street dimensions. The average, minimum and maximum street dimensions are presented in a new table added in the section "Setup for local-scale simulations". Regarding street length, these values are 179.3m, 3.0m (tunnels) and 1096.8m respectively.

Line 147 of the original version: "MUNICH assumes that the height and width of each street segment are constant, and that concentrations are uniform within the street segment." is replaced by: "MUNICH assumes that the height and width of each street segment are constant, and that concentrations are uniform within the street segment. Because MUNICH is a stand-alone model, it does not have any constraint on street dimensions. However, in the SinG model, street height cannot be higher than the first vertical level of the regional-scale module."

- Line 172: How is the standard deviation of the vertical wind speed computed from WRF variables?

Reply: Standard deviation of vertical wind is computed according to the atmospheric stability. Formulations and variables used for stable, neutral and unstable atmospheric conditions may be found in the paper of Soulhac et al. (2011). In line 172, the words "the standard deviation of the vertical wind speed" are replaced by "the standard deviation of the vertical wind speed, which are calculated depending on the atmospheric stability (Soulhac et al. 2011), .."

- Line 174: The CB05 is a gas-phase mechanism. Please, replace "concentration of

pollutants" for "concentration of gases" and "module" for "mechanism". Done.

- Line 184: In equation 9, is the parameter triangle sub-zero the same as triangle sub-one but for time n? Please, clarify the meaning of the notation used to define triangle sub one.

Reply: Equation (9) of the original version is corrected, and the times n and n+1 specified. Furthermore, the missing value of relative error precision (delta sub-zero) is added (it is equal to 0.01).

- Line 193: What is the computational overhead of running MUNICH coupled to Polair3D in SinG compared with running only Polair3D?

Reply: The increase of the computational time of running SinG compared to running only Polair3D or only MUNICH is of the order of a factor 1.28, using 100s of time-step and running both simulations in a machine with 256Go of RAM and processors Bi-Xeon E5-2650 v4 12 cores 2.2GHz. Note that MUNICH was not parallelized in the simulations performed here.

Line 128 of the original version: "This dynamic (two ways) coupling presents several advantages compared to a one-way formulation, as: (i) concentrations at the local and regional scales affect each other; (ii) no double counting of emissions is performed; (iii) the chemical and physical parametrizations used at the local and regional scales are consistent: both scales use the same chemical module and meteorological data. The regional and local-scale model, Polair3D and MUNICH, are now described emphasizing the numerical parameters and assumptions investigated in this study." is replaced by: "This two-way coupling presents several advantages compared to a one-way formulation, as: (i) concentrations at the local and regional scales affect each other; (ii) no double counting of emissions is performed; (iii) the chemical and physical parameterizations used at the local and regional scales are consistent: both scales use the same chemical module and meteorological data. But this approach also increases the computational time by a factor of about 1.28 (if MUNICH is not parallelized, as in

the simulations performed here). The regional and local-scale model, Polair3D and MUNICH, are now described emphasizing the numerical parameters and assumptions investigated in this study.

- Line 195: Clarify in the text that "cell i" is the cell of the regional model.

Reply: Line 198 of the original version: "Therefore, for each cell i, the background concentration over the canopy $C_{i,bg,cor}$ are obtained from regional-scale concentrations corrected to take into account the presence of buildings:" is replaced by: " Therefore, for each cell i of the regional model, the background concentrations over the canopy $C_{i,bg,cor}$ are obtained from regional-scale concentrations corrected to take into account the presence of buildings:"

- Line 203: From equation 11, SinG does not perform an average but a sum of the mass of background and the street. Please, amend. Why the authors use Vcell instead of (Vcell-Vbuild)? This is the exact volume from where equation 11 derives the mass.

Reply: Yes, the regional-scale masses are obtained by a sum, but the concentrations are obtained by an average. At the regional scale, the buildings and the streets are not taken into account. To be consistent with the regional-scale concentrations computed by Polair3D, the output regional-scale concentrations are taken equal to the mass average over the whole mesh. Note that the background concentrations used when computing the local-regional scale interactions are the regional-scale concentrations corrected by the presence of buildings, as detailed in equation (10).

- Line 206: I guess there is an error in the numbers of sub-Sections 2.4, 2.5, 2.6 and 2.7. sub-Section 2.4 should be a new Section 3, and the following sub-sections the new 3.1, 3.2 and 3.3. Done. Line 117 on original version: "The local, regional and multi-scale models MUNICH, Polair3D and SinG are presented in the first section of this paper. The second section describes the setup of the simulations over Paris city. The third section studies the impact of the stationary hypothesis and the numerical stability of the multi-scale model. The fourth section compares the simulated concentrations with air-quality measurements at traffic and background stations. Finally, the fifth section studies the influence of the dynamic coupling between the regional and local scales." is replaced by: "The local, regional and multi-scale models MUNICH, Polair3D and SinG are presented in the second section of this paper. The third section describes the setup of the simulations over Paris city. The fourth section studies the impact of the stationary hypothesis and the numerical stability of the multi-scale model. The fifth section compares the simulated concentrations with air-quality measurements at traffic and background stations. Finally, the sixth section studies the influence of the dynamic coupling between the regional and local scales."

- Line 208: Did the authors perform any spin-up in the chemistry?

Reply: We considered a spin-up of two days. Line 208 of the original version: "This sections describes the model configuration as well as the input data used for the regional and local-scale simulations. All simulations are performed from the 1st to 28th May 2014." is replaced by: "This section describes the model configuration as well as the input data used for the regional and local-scale simulations. All simulations are performed from the 1st to 28th May 2014, with a spin-up of two days."

- Line 210: Please clarify if SinG runs the 4 Polair3D domains or it is a decoupled run? Is Polair3D using two-way nesting?

Reply: Line 210 of the original version: "SinG is applied over Paris city, using a spatial resolution of 1 km × 1 km. obtained from one-way nesting simulations using Polair3D over three additional simulations covering Europe (domain 1), France (domain 2) and Île-de-France region (domain 3)." is replaced by: "The two-way SinG model is applied over Paris city (domain 4), using a spatial resolution of 1 km × 1 km. Initial and boundary conditions are obtained from one-way nesting simulations using Polair3D over three additional simulations covering Europe (domain 1), France (domain 2) and Île-de-France region (domain 3)."

- Line 240: Not many streets are used in the local-scale domain. What are the implications in the total traffic emissions of Paris ingested in the models then? Can the Authors quantify the percentage of emissions that the main streets used in the simulations represent from the total? How are the rest of the streets treated SinG during the two-way coupling? Are all the streets still used in eq 10 for Vbuild_i or only the main streets?

Reply: Within Paris, emissions in the streets of the street network represent most of the traffic emissions (94%). All the streets from the street network are used in equation (10) and considered in the two-way coupling. The rest of the streets are treated as surfacic emissions, and they are not involved in the two-way coupling.

- Line 268: What is the temporal resolution of the background concentrations used in MUNICH? This may explain part of the differences seen between SinG and MUNICH. SinG may use background conditions with higher temporal variability compared with MUNICH set up.

Reply: Polair3D, SinG and MUNICH simulations were performed using the same temporal resolutions for the background concentrations. Each MUNICH simulations employed the correspondent Polair3D results as background concentrations inlet.

Line 268 of the original version: "MUNICH simulations also require background concentrations as input data. They are obtained from a Polair3D simulation over the Paris city regional-scale domain. Note that the Polair3D simulation uses all emissions, including traffic, as input data (as indicated in Figure 4)." is replaced by: "MUNICH simulations also require background concentrations as input data. They are obtained from Polair3D simulations over the Paris city regional-scale domain. Note that the Polair3D simulations use all emissions, including traffic, as input data (as indicated in Figure 4), and that Polair3D, SinG and MUNICH simulations are performed using the same temporal resolution."

- Line 292: Why the authors consider that some of the results are not stable numerically? None of the runs shows numerical instabilities, at least from what can be seen

in Figure 7 and 8. The solution using 600s is quite similar to the one with 100s. Which are the criteria to identify numerical instabilities in SinG or MUNICH?

Reply: Considerable variations were observed on NO2 and NO concentrations in the streets after changing the simulation time step with the stationary approach (unlike the non-stationary approach). The paper is modified to quantify the numerical instabilities. Line 290 of the original version: "However, concentrations of NO2 and NO are highly dependent on the choice of the time step when the stationary hypothesis is made. This time-step dependency is observed using both MUNICH and This problem is solved with the non-stationary simulations, where concentrations of NO2 and NO are numerically stable and independent of the choice of the main time step." is replaced by: "However, in both MUNICH and SinG, street concentrations of NO2 and NO are highly dependent on the choice of the time step when the stationary approach is used. This problem is solved with the non-stationary simulations, where street concentrations of NO2 and NO are numerically stable and independent of the choice of the main time step. For example, regarding the concentrations simulated at CELES station by MUNICH with the stationary approach, the modification of the time step from 600s to 100s decreased by 5% NO2 concentrations and increased by 12% NO concentrations. With the non-stationary approach, these differences reduced to 0.1% for NO2 concentrations and 0.2% for NO concentrations."

- Line 293: Please, show the observations in Figure 7 and 8. Why do the Authors not show SinG and MUNICH results in the same station both figures? It is difficult to appreciate the differences between methodologies using different sites. Done. Line 287 on original version: "Figures 7 and 8 represent the time evolution of average daily concentrations of NOx , NO2 and NO during the simulation period, as simulated with MUNICH and SinG, at BONAP and CELES stations respectively." is replaced by "Figures 7 and 8 represent the time evolution of average daily concentrations of NOx , NO2 and NO during the simulation period, as simulated with MUNICH and SinG, at CELES station." Figure 8 on original version: "Daily-average concentrations of NOx

(left panel), NO2 (middle panel), and NO (right panel) concentrations calculated by SinG at BONAP station with different main time steps, using the stationary and non-stationary approaches." replaced by: "Figure 8. Daily-average concentrations of NOx (left panel), NO2 (middle panel), and NO (right panel) concentrations [$\mu$g.m$-3$] calculated by SinG at CELES station with different main time steps, using the stationary and non-stationary approaches."

- Line 295: Why a time step of 100s is selected with the non-stationary approach? Figure 7 and 8 show the same results with 600s, which imply that the model should be much faster with the same accuracy with 600s.

Reply: Line 290 of the original version: "However, concentrations of NO2 and NO are highly dependent on the choice of the time step when the stationary hypothesis is made. This time-step dependency is observed using both MUNICH and SinG. This problem is solved with the non-stationary simulations, where concentrations of NO2 and NO are numerically stable and independent of the choice of the main time step. Besides the numerical stability, NO2 and NO average concentrations obtained using the non-stationary approach are closer to observations than those using the stationary hypothesis, as indicated in Table 3. Therefore, in the rest of this paper only the simulations performed with the non-stationary approach and a main time step of 100 s are analyzed." is replaced by: "However, in both MUNICH and SinG, street concentrations of NO2 and NO are highly dependent on the choice of the time step when the stationary approach is used. This problem is solved with the non-stationary simulations, where street concentrations of NO2 and NO are numerically stable and independent of the choice of the main time step. For example, regarding the concentrations simulated at CELES station by MUNICH with the stationary approach, the modification of the time step from 600s to 100s decreased by 5% NO2 concentrations and increased by 12% NO concentrations. With the non-stationary approach, these differences reduced to 0.1% for NO2 concentrations and 0.2% for NO concentrations. Note that there are differences in the background concentrations of the regional-scale model if a time step

of 600 s is used rather than 100 s. This explains the small differences on NO2 concentrations observed at CELES station in Figure 8 using SinG with two different time steps (100 s and 600 s) and the non-stationary approach. Therefore, in the rest of this paper only the simulations performed with the non-stationary approach and a main time step of 100 s are analyzed. Besides the numerical stability, NO2 and NO average concentrations simulated using the non-stationary approach are closer to observations than those simulated using the stationary approach, as shown in Figures 7 and 8. The fraction bias of daily-average concentrations calculated with SinG (with a 100 s time-step) at CELES station is as high as 53% and -24% for NO2 and NO respectively using the stationary approach, and it is reduced to 13% and 4% respectively using the non-stationary approach."

Note that Table 3 of original version was reduced, as figures are now in the text.

- Line 304: Please, provide MUNICH results in Table 5 or clarify if MUNICH results are the same as the background concentration of Polair3D in open-areas.

Reply: MUNICH results can not be added to Table 5, because they are calculated only at the local-scale. Background concentrations used in MUNICH are those of Polair3D.

- Line 325: Figure 11 shows a better agreement of SinG and MUNICH during the morning peak than the evening one. Do the Authors have an explanation for this behaviour?

Reply: The following sentence is added line 326 of the original version: 'The better agreement of SinG and MUNICH during the morning peak than the evening one may be due to difficulties in modelling the atmospheric boundary height in the evening, and to higher day-to-day variability of traffic emissions in the evening than in the morning."

- Line 330: The Polair3D shows a substantial underestimation of NO as most CTMs. It appears a significant drawback for MUNICH and SinG to reproduce NO in open areas within the city. Can the Authors elaborate on approaches to overcome this limitation?

Reply: NO was strongly underestimated at stations located in big squares, such as

OPERA. Even if this underestimation was reduced using the non-stationary approach, the assumption of uniform concentration in these squares may not be adapted for NO, considering its short life time. The model could be improved by a better description of these squares with more accurate wind speed profiles and advection mass fluxes. Furthermore, the length of streets in MUNICH could be limited.

- Line 441: Results of SinG at the regional scale over the Paris area show higher NOx concentrations than Polair3D. What is the impact downwind Paris area in some urban background or rural sites using SinG or Polair3D? Does this increase of NOx in SinG results in a positive effect on the model downwind Paris (i.e., O3)? If this is the case, it would be relevant to elaborate on this because it has implications in the way how urban-cities are modeled in mesoscale models.

Reply: O3 background concentrations obtained with SinG are in average 5.90% larger than those obtained by Polair3D, with a maximal value of 20%. These relative differences of O3 concentrations have a similar spatial distribution as observed in Figure B2 (right panel), limited mainly inside Paris city. No considerable differences are observed outside the street-network.

- Line 452: As mentioned before, equation 11 does not define SinG regional scale concentration as an average but a sum of masses.

Yes, the regional-scale masses are obtained by a sum, but the concentrations are obtained by an average.

- Line 480: I suggest to add a last sentence highlighting that NO is less sensitive to this coupling and why.

Line 478 of the original version: "Although, on average over the streets of Paris, the influence of the dynamic coupling on NO2 concentrations in the street is only 7.5%, it can reach values as high as 63%. The influence of the dynamic coupling on background regional NO2 concentrations can be large as well: 11% on average over Paris with a

maximum relative difference of 34%." is replaced by: "Although, on average over the streets of Paris, the influence of the dynamic coupling on NO2 concentrations in the street is only 7.5%, it can reach values as high as 63%. The influence of the dynamic coupling on background regional NO2 concentrations can be large as well: 11% on average over Paris with a maximum relative difference of 34%. Because NO background concentrations are very low, and because of its short lifetime, NO concentrations are less sensitive to two-way dynamic coupling than NO2."

- Line 481: Do the Authors have any plan to evaluate other gases at street-level in the future? One of the most important capabilities of SinG is solving complex chemistry at street-scale. Understand the dynamics of other reactive gases in the urban environment deserves future research efforts.

Yes, this would be very interesting. At the moment, the limitations lie in the availability of street-scale measurements for comparisons.

Technical comments:

- Figure and Table captions: all captions should be self-explanatory. Several Tables and Figures present information that is not described in the caption (i.e., name of variables, units, the meaning of acronyms or abbreviations.) Figure 3: Original title: Domains simulated using WRF: Europe (D01), France (D02), Île-de-France region (D03), and Paris city (D04). Replaced by: Simulated domains using WRF: Europe (D01), France (D02), Île-de-France region (D03), and Paris city (D04).

Figure 4: Original title: For the Paris simulations using Polair3D and SinG, average anthropogenic emissions of NO2 [$\mu$g.s-1 .m-2] used as input of the regional-scale simulation with Polair3D (left panel), and as input of the regional-scale module of the multi-scale simulation with SinG (right panel).

Replaced by: Average over the simulation period of NO2 anthropogenic emissions [$\mu$g.s-1 .m-2 ] used as input of the regional-scale simulations over Paris city with Polair3D (left panel), and as input of the regional-scale module of the multi-scale simulations with SinG (right panel).

Figure 7: Original title: Daily-average concentrations of NOx (left panel), NO2 (middle panel), and NO (right panel) concentrations calculated by MUNICH at CELES station with different main time steps, using the stationary and non-stationary approaches. Replaced by: Daily-average concentrations of NOx (left panel), NO2 (middle panel), and NO (right panel) concentrations [$\mu$g.m-3] calculated by MUNICH at CELES station with different main time steps, using the stationary and non-stationary approaches.

Figure 8: Original title: Daily-average concentrations of NOx (left panel), NO2 (middle panel), and NO (right panel) concentrations calculated by SinG at BONAP station with different main time steps, using the stationary and non-stationary approaches. Replaced by: Daily-average concentrations of NOx (left panel), NO2 (middle panel), and NO (right panel) concentrations [$\mu$g.m-3] calculated by SinG at CELES station with different main time steps, using the stationary and non-stationary approaches.

Table 4 of the original version: Original title: Statistics at traffic stations (o and s represent the average observed and simulated concentrations respectively). Replaced by: Statistics at traffic stations (o and s represent the average observed and simulated concentrations respectively, in $\mu$g.m-3).

Table 5 of the original version: Original title: Statistics at background stations (o and s represent the average observed and simulated concentrations respectively. Replaced by: Statistics at background stations (o and s represent the average observed and simulated concentrations respectively, in $\mu$g.m-3.

Figure 9: Original title: Daily-average concentrations of NOx (left panel), NO2 (middle panel), and NO (right panel) concentrations observed and simulated at CELES station with MUNICH, SinG and Polair3D Replaced by: Daily-average concentrations of NOx (left panel), NO2 (middle panel), and NO (right panel) concentrations [$\mu$g.m-3] observed and simulated at CELES station with MUNICH, SinG and Polair3D.

Figure 10: Original title: Daily-average concentrations of NOx (left panel), NO2 (middle panel), and NO (right panel) concentrations observed and simulated at SOULT station with MUNICH, SinG and Polair3D Replaced by: Daily-average concentrations of NOx (left panel), NO2 (middle panel), and NO (right panel) concentrations [$\mu$g.m-3] observed and simulated at SOULT station with MUNICH, SinG and Polair3D.

Figure 11: Original title: Hourly-average concentrations of NOx (left panel), NO2 (middle panel), and NO (right panel) concentrations observed and simulated at SOULT station with MUNICH, SinG and Polair3D. Replaced by: Hourly-average concentrations of NOx (left panel), NO2 (middle panel), and NO (right panel) concentrations [$\mu$g.m-3] observed and simulated at SOULT station with MUNICH, SinG and Polair3D.

Table 6 of the original version: Original title: Average concentrations measured and simulated with SinG of NOx , NO2 , NO and NO2 /NO ratios at traffic stations (o and s represent the observed and simulated average respectively). Replaced by: Average concentrations measured and simulated with SinG of NOx , NO2 , NO and NO2 /NO ratios at traffic stations (o and s represent the observed and simulated average respectively, in $\mu$g.m-3).

Figure 12: Original title: Daily concentrations of NOx (left panel), NO2 (middle panel), and NO (right panel) concentrations observed and simulated at PA04C station with SinG and Polair3D. Replaced by: Daily-average concentrations of NOx (left panel), NO2 (middle panel), and NO (right panel) concentrations [$\mu$g.m-3] observed and simulated at PA04C station with MUNICH, SinG and Polair3D.

Figure 13: Original title: Daily concentrations of NOx (left panel), NO2 (middle panel), and NO (right panel) concentrations observed and simulated at PA13 station with SinG and Polair3D. Replaced by: Daily-average concentrations of NOx (left panel), NO2 (middle panel), and NO (right panel) concentrations [$\mu$g.m-3] observed and simulated at PA13 station with MUNICH, SinG and Polair3D.

Table 7 of the original version: Original title: Street characteristics at traffic stations.

Replaced by: Street length (L), aspect ratio (alpha_r), number of connected streets, and the correspondent relative difference of NO2 concentrations calculated by SinG and MUNICH at each traffic station.

Figure 15: Original title: NO2 daily concentrations in the street and in the background at CELES traffic station. Replaced by: NO2 daily-average concentrations [$\mu$g.m-3] in the street and in the background at CELES traffic station.

Figure 16: Original title: Daily weighted mass fluxes of NO2 at BONAP (left panel), CELES (middle panel) and BP_EST (right panel) traffic stations. Replaced by: Daily-weighted mass fluxes of NO2 at BONAP (left panel), CELES (middle panel) and BP_EST (right panel) traffic stations.

Figure 17: Original title: Daily weighted mass flux of NO at BONAP (left panel), CELES (middle panel) and BP_EST (right panel) traffic stations. Replaced by: Daily-weighted mass flux of NO at BONAP (left panel), CELES (middle panel) and BP_EST (right panel) traffic stations.

Figure 18: Original title: Percentage of streets present in each $\alpha$r interval according to $\alpha$r values and the NO2 (left panel) and NO (right panel) relative differences between pollutant concentrations calculated by SinG and MUNICH. Replaced by: Percentage of streets (purple color) present in each $\alpha$r interval according to $\alpha$r values and the NO2 (left panel) and NO (right panel) relative differences between pollutant concentrations calculated by SinG and MUNICH.

- Equations: There are several equations with the definition of terms in the same line. Please, split those cases in separate equations. This occurs in Eq 3, 7, 8, 9, 11. Done.

- Line 2: Delete "i.e." Done.

- Line 15: Amend the format of the World Health Organization reference. Use "WHO (2016)" or "World Health Organization (2016)". It has not much sense to define an acronym that will not be used anymore in the text. Done.

- Line 25: Add a comma after "e.g." or "i.e.". Done.

- Line 25: Follow the appropriate style used by ACP when using references between parentheses. To simplify the text, I suggest using only the references, not the model acronyms.

Line 25 of the original version: "Regional-scale chemistry-transport models (CTMs), as three-dimension gridded Eulerian models (e.g. Polair3D (Sartelet et al., 2007), WRF-Chem (Zhang et al., 2010), CHIMERE (Menut et al., 2014), CMAQ (Community Multi-scale Air Quality Modeling System) (Byun and Ching, 1999), AURORA (Mensink et al., 2001)) solve a chemistry-transport equation for chemical compounds or surrogates, taking into account pollutant emissions, transport (advection by winds, turbulent diffusion), chemical transformations, and dry/wet depositions." is replaced by: "Regional-scale chemistry-transport models (CTMs), as three-dimension gridded Eulerian models solve a chemistry-transport equation for chemical compounds or surrogates, taking into account pollutant emissions, transport (advection by winds, turbulent diffusion), chemical transformations, and dry/wet depositions. Several CTMs are available in the literature, e.g., Polair3D, WRF-Chem, CHIMERE, Community Multi-scale Air Quality Modeling System (CMAQ), Air Quality Model For Urban Regions Using An Optimal Resolution Approach (AURORA), described in Sartelet et al. (2007); Zhang et al. (2010); Menut et al. (2014); Byun and Ching (1999); Mensink et al. (2001) respectively."

- Line 44: Use the same style to introduce acronyms throughout the text, first the complete name followed by the acronym in parentheses.

Line 6 on original version: "This coupling combines the regional-scale chemistry-transport model Polair3D and the street network model MUNICH (Model of Urban Network of Intersecting Canyons and Highway)." replaced by: "This coupling combines the regional-scale chemistry-transport model Polair3D and the street network model Model of Urban Network of Intersecting Canyons and Highway (MUNICH) with

a two-way feedback." Line 125 on original version: "the street-network model MUNICH (Model of Urban Network of Intersecting Canyons and Highways)." replaced by "the street-network model Model of Urban Network of Intersecting Canyons and Highways (MUNICH)". Line 216 on original version: "The initial and boundary conditions of the largest domain (over Europe) are obtained from a global-scale chemical-transport simulation using MOZART-4 (model for Ozone and Related Chemical Tracers) (Emmons et al., 2010) coupled to the aerosol module GEOS-5 (Goddard Earth Observing System Model) (Chin et al., 2002). The spatial resolution of the MOZART- 4/GEOS-5 simulation is 1.9◦ × 2.5◦ , with 56 vertical levels." is replaced by: "The initial and boundary conditions of the largest domain (over Europe) are obtained from a global-scale chemical-transport simulation using the Model for Ozone and Related Chemical Tracers (MOZART-4) coupled to the aerosol module Goddard Earth Observing System Model (GEOS-5), described in Emmons et al. (2010) and Chin et al. (2002), respectively. The spatial resolution of the MOZART-4/GEOS-5 simulation is 1.9◦ × 2.5◦ , with 56 vertical levels."

- Line 44: Avoid opening a parenthesis just after a closing one, and use similar style format if several references are provided. In particular, "(CALINE4; Bensons, 1984; Sharma, et al.). Check and unify the reference format used in the text.

Line 44 of the original version: "Afterward, street-network models, such as CALINE4 (California Line source dispersion model) (Benson, 1984), (Sharma et al.) and CAR (Calculation of Air pollution from Road traffic model) (Eerens et al., 1993), assume that pollutant dispersion follows a Gaussian plume distribution and traffic emissions are line sources. Other models expanded this formulation combining a Gaussian plume and a box model, e.g. CPBM (Canyon Plume Box Model) (Yamartino and Wiegand, 1986), OSPM (Operational Street Pollution Model) (Berkowicz et al., 1997; Berkowicz, 2000), and ADMS-Urban (Atmospheric Dispersion Modeling System) (McHugh et al., 1997). The Gaussian plume model is used to estimate the direct contribution of traffic emissions, and the box model calculates the recirculation contribution, resultant from the

wind vortex formed in the street canyon." Is replaced by: "Other street-network models assume that pollutant dispersion follows a Gaussian plume distribution and consider traffic emissions as line sources, as the Calculation of Air pollution from Road traffic model (CAR) and the California Line source dispersion model (CALINE4), developed by Eerens et al. (1993) and Sharma et al. (2013) respectively. Other models expanded this formulation combining a Gaussian plume and a box model, e.g., the Canyon Plume Box Model (CPBM), the Operational Street Pollution Model (OSPM), and the urban version of Atmospheric Dispersion Modeling System (ADMS-Urban). The Gaussian plume model is used to estimate the direct contribution of traffic emissions, and the box model calculates the recirculation contribution, resultant from the wind vortex formed in the street canyon (Yamartino and Wiegand, 1986; Berkowicz et al., 1997; Berkowicz, 2000; McHugh et al., 1997)." Line 65 in original version: "The Model of Urban Network of Intersecting Canyons and Highways (MUNICH) (Kim et al., 2018) presents a similar box-model parameterization as SIRANE, but it does not employ a Gaussian model to determinate background concentrations." is replaced by: "The Model of Urban Network of Intersecting Canyons and Highways (MUNICH), developed by Kim et al. (2018), presents a similar box-model parameterization as SIRANE, but it does not employ a Gaussian model to determinate background concentrations."

- Line 44: The reference "Sharma et al." is incomplete. Please, provide the year here and in the reference section. Done.

Line 612 of the original version: "Sharma, N., Gulia, S., Dhyani, R., and Singh, A.: Performance evaluation of CALINE 4 dispersion model for an urban highway corridor in Delhi, J. Sci. Ind. Res."is replaced by: "Sharma, N., Gulia, S., Dhyani, R., and Singh, A.: Performance evaluation of CALINE 4 dispersion model for an urban highway corridor in Delhi, J. Sci. Ind. Res., 72, 521–530, 2013."

- Line 52: Define the acronym SIRANE, as done previously with other models.

There is no definition of the acronym of SIRANE in the literature.

- Line 136: Amend the use of references. In this case should be like "at different locations (e.g., Sartelet et al., 2012; Abdallah et al., 2018; ...)." Done.

- Line 137: Remove "including Greater Paris" from the middle of the list of references. Done.

Line 136 of the original version: "Polair3D was used in many studies to simulate gas and particle concentrations at regional scale at different locations, e.g. Sartelet et al. (2012), Abdallah et al. (2018), including Greater Paris Sartelet et al. (2018), Zhu et al. (2016a), Zhu et al. (2016b), Kim et al. (2015), Kim et al. (2014), Couvidat et al. (2013), Royer et al. (2011)." is replaced by: "Polair3D was used in many studies to simulate gas and particle concentrations at regional scale at different locations (e.g., Royer et al. (2011), Sartelet et al. (2012), Couvidat et al. (2013), Kim et al. (2014), Kim et al. (2015), Zhu et al. (2016a), Zhu et al. (2016b), Abdallah et al. (2018), Sartelet et al. (2018))."

- Line 150, Equation 2: The use of Q and Qinflow are confusing. Q is a mass variable while Qinflow, Qemis, Qoutflow, ..., Qdep are mass fluxes. I suggest using another letter for the fluxes, e.g., Finflow.

For mass, the term Q is replaced by M.

- Line 153: Amend "each of this term" with "each term". Done

- Line 158: Please, define what is Qair and Cst like is done with H, W and Ust.

Line 158 of the original version: "where H and W are the street height and width, and ust is the mean air velocity in the street," is replaced by: "where Qair is the air flow, Cst the pollutant concentration in the street, H and W are the street height and width, and ust the mean air velocity in the street,"

- Line 161: Together with the reference of equation 8 from Kim et al. (2018), add the reference to equation 7 of this manuscript. Done.

Line 161 of the original version: "According to the equation (8) of Kim et al. (2018), Qvert is inversely proportional to the aspect ratio $\alpha r$ of the street." is replaced by: "According to the equation (8) of Kim et al. (2018) and equation 8 of this paper, Qvert is inversely proportional to the aspect ratio $\alpha r$ of the street."

- Line 194: Amend "details" with "detailed". Done.

- Line 220: Specify the version of WRF model used. Done.

Line 220 of the original version: "Meteorological data for the four domains are calculated by the WRF model (Weather Research and Forecasting) (Skamarock et al., 2008) with a two-way nesting" is replaced by: "Meteorological data for the four domains are calculated by the model Weather Research and Forecasting (WRF) version 3.9.1.1 with a two-way nesting (Skamarock et al., 2008)"

- Line 222: Is the top of the atmosphere in WRF set at 5000m? If not, amend.

No, the top of atmosphere in WRF simulations is 21000m.

Line 222 of the original version: "45 km, 9 km $\times$ 9 km, 3 km $\times$ 3 km and 1 km $\times$ 1 km for domains 4 to 1 respectively), with 38 vertical levels, from 0 to 5000 m." is replaced by: "45 km, 9 km $\times$ 9 km, 3 km $\times$ 3 km and 1 km $\times$ 1 km for domains 4 to 1 respectively), with 38 vertical levels, from 0 to 21km."

- Line 225: Delete "and chemical". Done.

- Line 228: Specify the version of MEGAN model used.

Line 227 of the original version: "Biogenic emissions over all domains are estimated using the Model of Emissions of Gases and Aerosols from Nature (MEGAN)." is replaced by: "Biogenic emissions over all domains are estimated using the Model of Emissions of Gases and Aerosols from Nature (MEGAN v2.04)."

- Line 233: Amend the link provided. It is not working.

Line 233 of the original version: https://trimis.ec.europa.eu/sites/default/files/project/documents/20090917_162316_73833_
%20Final%20Report.pdf is replaced by: https://trimis.ec.europa.eu/project/healthier-
environment-through-abatement-vehicle-emission-and-noise

- Line 236: Specify the period of average used in Figure 4.

Figure 4Âǎ: Original title: For the Paris simulations using Polair3D and SinG, aver-
age anthropogenic emissions of NO2 [$\mu$g.s-1 .m-2] used as input of the regional-scale
simulation with Polair3D (left panel), and as input of the regional-scale module of the
multi-scale simulation with SinG (right panel). Replaced by: Average over the sim-
ulation period of NO2 anthropogenic emissions [$\mu$g.s-1 .m-2 ] used as input of the
regional-scale simulations over Paris city with Polair3D (left panel), and as input of the
regional-scale module of the multi-scale simulations with SinG (right panel).

- Line 284: Figure 6 and Figure 5 could be combined in a single figure. I recommend
using two different colours to differentiate urban and traffic measurement stations (i.e.,
red and black dots).

We performed the modifications proposed in Figure 6, but both figures were kept in the
manuscript to maintain the order of sections. Line 279 on original version: "Simulated
concentrations are compared with air-quality measurements at traffic and urban back-
ground stations. Figure 6 represents the street network used in this study, displaying
the regional-scale grid mesh and the position of all stations con- sidered. Air-quality
stations comprise 5 urban stations (indicated by PA04C, PA07, PA12, PA13 PA18),
and 8 traffic stations (BONAP, ELYS, HAUSS, CELES, BASCH, OPERA, SOULT and
BP_EST)." is replaced by: "Simulated concentrations are compared with air-quality
measurements at traffic and urban background stations. Figure 6 represents the street
network emissions used in this study (see section 3.2), also displaying the regional-
scale grid mesh and the position of all stations considered. Air-quality stations com-
prise 5 urban stations (indicated by PA04C, PA07, PA12, PA13 PA18, with blue dots),
and 8 traffic stations (BONAP, ELYS, HAUSS, CELES, BASCH, OPERA, SOULT and
BP_EST, with red dots).". Also, the indication of stations located in high emission streets and/or adjacent to big squares are indicated in Table 6 (modified from the original version), as proposed by the first referee.

- Line 358: Please, clarify how relative difference is computed. Which is the reference value?

The reference concentrations at the regional scale are Polair3D concentrations, and at the local scale, MUNICH concentrations.

Line 358 of the original version: "Figure 14 represents the mean relative differences between NO2 concentrations simulated using coupled and non-coupled simulations at local (differences between SinG and MUNICH) and regional scales (differences between SinG and Polair3D), averaged over the simulation period. In average, these mean relative differences are about 7.5% at the local scale and 11.3% at the regional scale." is replaced by: "Figure 14 represents the mean relative differences between NO2 concentrations simulated using coupled and non-coupled simulations at local (differences between SinG and MUNICH) and regional scales (differences between SinG and Polair3D), averaged over the simulation period. In average, these mean relative differences are about 7.5% at the local scale and 11.3% at the regional scale. To compute these relative differences, MUNICH and Polair3D concentrations were adopted as reference concentrations at the local and regional scales, respectively."

- Line 361: It would be desirable to use the same range and colour scale in both panels of Figure 14. We changed the color scale such as being having similar color for similar percentage.

- Line 389: Amend "daily mass fluxes" with "daily weighted mass fluxes". Done.

- Line 396: In the legends of Figure 16, use the same notation as equation 12. Instead of Qinflow should be qf_inflow. Done.

- Line 401: Check the use of the hyphen. Sometimes is used and others not (e.g., daily

weighted, daily-weighted). Done.

- Line 439: Detailed the meaning of the colour in the Figure by adding "Percentage of streets (purple colour). Add a % at the top of the colour scale. Done.

- Line 460: Re-word "coupling finely".

Line 460 of the original version: "A non-stationary dynamic approach coupling finely chemistry and transport of pollutants was implemented and proved to be numerically stable. It leads to NO2 and NOx concentrations that compare well to observations, both at the regional and local scales." is replaced by: "A non-stationary dynamic approach was implemented, by solving with a second order numerical scheme the transport of pollutants and chemistry. This approach proved to be numerically stable, with a good agreement between observed and simulated concentration of NO2 and NOx at both regional and local scales."

- Line 467: Use "appropriate" instead of "verified".

Line 466 of the original version: "This underestimation is probably due to the short life time of NO, for which the assumption of uniform concentrations in wide streets and big squares may not be verified." is replaced by: "This underestimation is probably due to the short lifetime of NO, for which the assumption of uniform concentrations in wide streets and big squares may not be appropriate."

- Line 497: Correlation is never used in the manuscript. Please, remove the statistic from the Annex. Done.

Please also note the supplement to this comment:
https://www.atmos-chem-phys-discuss.net/acp-2019-1087/acp-2019-1087-AC2-supplement.pdf
* * *
[Figure]

**Supplement:**

[revised manuscript text omitted]

---

## Editor Decision (ED1)

**Editor comments: MS acp-2020-1087**

Thank you for addressing most of the reviewer comments. However, I think, while your response to the reviewer comments was quite extensive, several of the responses could also be implemented better in the text as they might be also of interest to the readers of your paper. In addition, I have some further technical/minor comments that should be addressed.
In responding to my comments, please list them in a single document together with your responses, the line numbers of the revised manuscript with the changes and the marked-up manuscript.

**I. Referee & Editor comments**

**1) Reviewer #2 comment:** Finally, some discussion about the impact of the Street-in-Grid at the regional scale downwind the city is missing. It is clear that the two-way coupling will improve the skills of SinG at the regional scale if it is evaluated with urban sites, but does this result also in an improvement of the mesoscale model photochemistry downwind Paris? Is there any sensitivity in NOx and other reactive gases like O3 in some rural areas affected by the pollution plume of Paris?

**Author Response:** O3 background concentrations obtained with SinG are in average 5.90% larger than those obtained by Polair3D, with a maximal value of 20%. These relative differences of O3 concentrations have a similar spatial distribution as observed in Figure B2 (right panel), limited mainly inside Paris city. No considerable differences are observed outside the street-network.

**Editor comment:** Please add some information to the text that describes the small differences on the downwind side of Paris

2) **Reviewer #1 comment:** The authors should justify why only a relatively short period (1-28 May, 2014) is used for model validation.

**Author Response:** This paper aims at analyzing the influence of the non-stationary regime and multi-scale coupling at both local and regional scales. Many runs were performed for this sensitivity study, and a one-month simulation period is long enough to analyze the processes.

**Editor comment:** This information should be added at the beginning of Section 5.

**Author Response:** There is a mismatch between the year of emissions over Île-de-France of the domain 3 and over the domain 4 that is from 2012 and the model validation period of 2014.
Explain how this may influence comparison of model results and measurements.

**3) Reviewer #1 comment:** As specified in the paper, 2012 Airparif inventory is used only for sectors different than road traffic. Traffic emissions use data specific of 2013 and 2014. Comparisons between the 2012 Airparif inventory and the more recent 2015 Airparif inventory show that the most important differences in NOx emissions between the two years are due to differences in traffic emissions. Because traffic emissions are specific of the year studied here, we do not think that using the 2012 inventory for sources other than traffic impact our comparison of model results to measurements.

**Editor comment:** Please add also this information to the paper.

**4) Reviewer #2:** Figure and Table captions: all captions should be self-explanatory. Several Tables and Figures present information that is not described in the caption (i.e., name of variables, units, the meaning of acronyms or abbreviations.)

**Editor comment:** While I appreciate that you added units etc to the figure captions where appropriate, some of the captions and figures should be improved:

> **Figure 1:**
> 1) This caption needs more information: Add the models you used, and details on the simulation
> 2) Add a scale, either in latitude/longitude or km to the figures.
>
> **Figure 2,** caption: This caption is not self-explanatory. Please add more details on which simulations, models etc so that the reader understands what information for which reason
>
> **Figure 3:**
> 1) This caption needs more information: Add the models you used, and details on the simulation
> 2) Add a scale, either in latitude/longitude or km to the figure.
>
> **Figure 15, caption:** Add details on the simulation
>
> Figure 16: Improve the figure legend and clarify, either in the legend or caption, 'inflow, emis, vert, outflow'. Refer to Eq.-18 in the caption.
>
> **Table 3, caption:** Add more information and define the abbreviated words
>
> **Tables 4 and 5, caption:**
> - Not all readers might be familiar with all of the abbreviated statistical parameters. Define them in the caption (or in footnotes).
> - Add enough information that the reader can understand what simulations and assumptions these numbers refer to.
> - Instead of 'statistics', it should rather read 'statistical parameter' or 'statistical measures' or similar.

**II. Technical/minor editor comments:**

Line numbers refer to uploaded revised manuscript without annotations

**l. 128, here and in the remainder of the manuscript:** There is no need to repeat the definition of MUNICH

**l. 135:** replace 'model' by 'models'

**l. 170:** This reads as if you were referring to two different equations but they seem to be the same. I suggest writing: 'According to Eq.8. …'
and then at Eqs.8 and 9, you add the reference to Kim et al. 2018.

**l. 183:** replace 'which are calculated' by 'which is calculated' – unless you also calculated the value of beta, when then deserves an extra sentence.

**l. 262:** Please make sure that you include the full name of the website. Currently it seems truncated and the link is not thus not working

**l. 334:** What are these 'performance criteria'? Are they defined somewhere? If so, please refer to the respective section; if not, add their definition.

**Table 7,** caption: Replace 'correspondent' by 'corresponding'

**l. 444:** 'concentrations' misspelled

---

## Author Response (AR2)

*Editor comments: MS acp-2020-1087*

*Thank you for addressing most of the reviewer comments. However, I think, while your response to the reviewer comments was quite extensive, several of the responses could also be implemented better in the text as they might be also of interest to the readers of your paper. In addition, I have some further technical/minor comments that should be addressed. In responding to my comments, please list them in a single document together with your responses, the line numbers of the revised manuscript with the changes and the marked-up manuscript.*

*I. Referee & Editor comments*

*1) Reviewer #2 comment: Finally, some discussion about the impact of the Street-in-Grid at the regional scale downwind the city is missing. It is clear that the two-way coupling will improve the skills of SinG at the regional scale if it is evaluated with urban sites, but does this result also in an improvement of the mesoscale model photochemistry downwind Paris? Is there any sensitivity in NOx and other reactive gases like O3 in some rural areas affected by the pollution plume of Paris?*

*Author Response: O3 background concentrations obtained with SinG are in average 5.90% larger than those obtained by Polair3D, with a maximal value of 20%. These relative differences of O3 concentrations have a similar spatial distribution as observed in Figure B2 (right panel), limited mainly inside Paris city. No considerable differences are observed outside the street-network.*

*Editor comment: Please add some information to the text that describes the small differences on the downwind side of Paris*

**Author response**: Line 484 "This justifies the higher differences between coupled and non-coupled simulations at the regional scale than at the local scale."
is replaced by "This justifies the higher differences between coupled and non-coupled simulations at the regional scale than at the local scale. Regarding the downwind side of Paris, the concentrations of NOx and $O_3$ simulated by Polair3D and SinG are similar outside the street-network region."

*2) Reviewer #1 comment: The authors should justify why only a relatively short period (1-28 May, 2014) is used for model validation.*

*Author Response: This paper aims at analyzing the influence of the non-stationary regime and multi-scale coupling at both local and regional scales. Many runs were performed for this sensitivity study, and a one-month simulation period is long enough to analyze the processes.*

*Editor comment: This information should be added at the beginning of Section 5.*

**Author response:** This information was added in section 3, before detailing the numerical stability results and the comparisons with air-quality measurements.
Line 224 "This section describes the model configuration as well as the input data used for the regional and local-scale simulations. All simulations are performed from the 1st to 28th May 2014, with a spin-up of two days."
is replaced by "This section describes the model configuration as well as the input data used for the regional and local-scale simulations. All simulations are performed from the 1st to 28th May 2014, with a spin-up of two days. A one-month simulation period is considered long enough to analyze the influence of the non-stationary regime and the multi-scale coupling between local and regional scales."

*3) Reviewer #1 comment: There is a mismatch between the year of emissions over Île-de-France of the domain 3 and over the domain 4 that is from 2012 and the model validation period of 2014. Explain how this may influence comparison of model results and measurements.*

*Author Response: As specified in the paper, 2012 Airparif inventory is used only for sectors different than road traffic. Traffic emissions use data specific of 2013 and 2014. Comparisons between the 2012 Airparif inventory and the more recent 2015 Airparif inventory show that the most important differences in NOx emissions between the two years are due to differences in traffic emissions. Because traffic emissions are specific of the year studied here, we do not think that using the 2012 inventory for sources other than traffic impact our comparison of model results to measurements.*

*Editor comment: Please add also this information to the paper.*

**Author response:** Line 248 "Over Île-de-France of the domain 3 and over the domain 4, they are calculated using the emission inventory of 2012, provided by the air-quality agency of Paris (AIRPARIF). For traffic emissions, AIRPARIF used the HEAVEN bottom-up traffic emissions model (https://trimis.ec.europa.eu/project/healthier-environment-through-abatement-vehicle-emission-and-noise) with fleet and technology data specific of 2013 and 2014. Anthropogenic emissions followed the vertical distribution defined by Bieser et al. (2011) for the different activity sectors. More details on emission data and speciations may be found in Sartelet et al. (2018)."
is replaced by "Over Île-de-France of the domain 3 and over the domain 4, they are calculated using the emission inventory of 2012, provided by the air-quality agency of Paris (AIRPARIF). Comparisons between the 2012 Airparif inventory and the more recent 2015 Airparif inventory show that largest differences in NOx emissions between the two years are due to differences in traffic emissions. For traffic emissions, fleet and technology data specific of 2013 and 2014 are used, and emissions are computed with the HEAVEN bottom-up traffic emissions model by Airparif (https://trimis.ec.europa.eu/project/healthier-environment-through-abatement-vehicle-emission-and-noise). Anthropogenic emissions followed the vertical distribution defined by Bieser et al. (2011) for the different activity sectors. More details on emission data and speciations may be found in Sartelet et al. (2018)."

*4) Reviewer #2: Figure and Table captions: all captions should be self-explanatory. Several Tables and Figures present information that is not described in the caption (i.e., name of variables, units, the meaning of acronyms or abbreviations.)*

*Editor comment: While I appreciate that you added units etc to the figure captions where appropriate, some of the captions and figures should be improved:*
*Figure 1:*
*1) This caption needs more information: Add the models you used, and details on the simulation*
*2) Add a scale, either in latitude/longitude or km to the figures.*

**Author response:** "Figure 1. Domains simulated: Europe (domain 1), France (domain 2), Île de France region (domain 3), and Paris city (domain 4)."
replaced by "Figure 1. Regional-scale domains: Europe (domain 1, with a spatial resolution of 45 km x 45 km), France (domain 2, with a spatial resolution 9 km x 9 km) and Île-de-France region (domain 3, with a spatial resolution 3 km x 3 km) for one-way nesting simulations using Polair3D, and Paris city (domain 4, with a spatial resolution of 1 km x 1 km) for simulations with SinG."
The scale was added in the Figure, as requested.

*Figure 2, caption: This caption is not self-explanatory. Please add more details on which simulations, models etc so that the reader understands what information for which reason*

**Author response:** "Figure 2. Vertical levels used in all regional-scale simulations."
is replaced by "Figure 2. Vertical levels used in all regional-scale simulations performed with Polair3D and SinG."

*Figure 3:*
*1) This caption needs more information: Add the models you used, and details on the simulation*
*2) Add a scale, either in latitude/longitude or km to the figure.*

**Author response:** "Figure 3. Simulated domains using WRF: Europe (D01), France (D02), Île-de-France region (D03), and Paris city (D04)."
is replaced by "Figure 3. Simulated domains using WRF to calculate meteorological data: Europe (D01, with a spatial resolution of 45 km x 45 km), France (D02, with a spatial resolution of 9 km x 9 km), Île-de-France region (D03, with a spatial resolution of 3 km x 3 km), and Paris city (D04, with a spatial resolution of 1 km x 1 km)."
The scale was added in the Figure, as requested.

*Figure 15, caption: Add details on the simulation*

**Author response:** "Figure 15. $NO_2$ daily-average concentrations [$\mu g.m^{-3}$] in the street and in the background at CELES traffic station."
is replaced by "Figure 15. $NO_2$ daily-average concentrations [$\mu g.m^{-3}$] in the street and in the background, using MUNICH (one-way dynamic coupling) and SinG (two-way dynamic coupling) at CELES traffic station."

*Figure 16: Improve the figure legend and clarify, either in the legend or caption, 'inflow, emis, vert, outflow'. Refer to Eq.-18 in the caption.*

**Author response:** "Figure 16. Daily-weighted mass fluxes of $NO_2$ at BONAP (left panel), CELES (middle panel) and BP_EST (right panel) traffic stations."
is replaced by "Figure 16. Normalized daily-weighted mass fluxes of $NO_2$ at BONAP (left panel), CELES (middle panel) and BP_EST (right panel) traffic stations: inflow advection mass-flux (qf_inflow) in blue, emission (qf_emis) in green, vertical mass-flux (qf_vert) in red, and outflow advection mass-flux (qf_outflow) in cyan. The fluxes are calculated as detailed in Equation (18)."

*Table 3, caption: Add more information and define the abbreviated words*

**Author response:** The abbreviations were removed and replaced by the complete words.
"Table 3. List of the sensitivity simulations performed"
is replaced by "Table 3. List of the sensitivity simulations performed, using both MUNICH and SinG with different time steps (100s and 600s), adopting or not the stationary hypothesis."

*Tables 4 and 5, caption: - Not all readers might be familiar with all of the abbreviated statistical parameters. Define them in the caption (or in footnotes). - Add enough information that the reader can understand what simulations and assumptions these numbers refer to. - Instead of 'statistics', it should rather read 'statistical parameter' or 'statistical measures' or similar.*

**Author response:** The definition of each statistical parameter is presented in Appendix A1. As recommended, a footnote is also added to improve clarity. For both figure captions, the term statistics is replaced by statistical parameters, as indicated below.

"Table 4. Statistics at traffic stations (o and s represent the average observed and simulated concentrations respectively, in $\mu g.m^{-3}$)."
is replaced by "Table 4. Statistical parameters[1] at traffic stations (o and s represent the average observed and simulated concentrations respectively, in $\mu g.m^{-3}$)."

"Table 5. Statistics at background stations (o and s represent the average observed and simulated concentrations respectively, in $\mu g.m^{-3}$)."
is replaced by "Table 5. Statistical parameters[1] at background stations (o and s represent the average observed and simulated concentrations respectively, in $\mu g.m^{-3}$)."

Footnote included: [1]FB represents the fractional bias, MG the geometric mean bias, NMSE the normalized mean square error, VG the geometric variance, NAD the normalised absolute difference, and FAC2 the fraction of predictions within a factor two of observations. They are calculated as detailed in Appendix A1.

***II. Technical/minor editor comments:*** *Line numbers refer to uploaded revised manuscript without annotations*

*l. 128, here and in the remainder of the manuscript: There is no need to repeat the definition of MUNICH*
**Author response:**
Line 127 "Street-in-Grid (SinG) is a multi-scale model that couples the street-network Model of Urban Network of Intersecting Canyons and Highways (MUNICH) with the 3D chemistry-transport model Polair3D using a two-way dynamic multi-scale approach."
is replaced by: "Street-in-Grid (SinG) is a multi-scale model that couples the street-network model MUNICH with the 3D chemistry-transport model Polair3D using a two-way dynamic multi-scale approach."

*l. 135: replace 'model' by 'models'*
**Author response:**
Line 135: "The regional and local-scale model," replaced by "The regional and local-scale models,"

*l. 170: This reads as if you were referring to two different equations but they seem to be the same. I suggest writing: 'According to Eq.8. …' and then at Eqs.8 and 9, you add the reference to Kim et al. 2018.*
**Author response**: Line 170 "According to the equation (8) of Kim et al. (2018) and equation (8) of this paper, Qvert is inversely proportional to the aspect ratio αr of the street. Therefore, the vertical mass transfer is more significant for wide streets than for street canyons."
is replaced by: "According to the equations (8) and (9), Qvert is inversely proportional to the aspect ratio αr of the street. Therefore, the vertical mass transfer is more significant for wide streets than for street canyons." The reference to Kim et al. (2018) is added where equations 8 and 9 are detailed.

*l. 183: replace 'which are calculated' by 'which is calculated' – unless you also calculated the value of beta, when then deserves an extra sentence.*
**Author response:** Line 183 "which are calculated depending on the" is replaced by "which is calculated depending on the".

*l. 262: Please make sure that you include the full name of the website. Currently it seems truncated and the link is not thus not working*
**Author response:** Done.

*l. 334: What are these 'performance criteria'? Are they defined somewhere? If so, please refer to the respective section; if not, add their definition.*
**Author response**: The performance criteria applied are defined in the beginning of section 5 (line 330), and they were proposed by Hanna and Chang (2012) and Herring and Huq (2018). To improve clarity, line 334: "indicators of Table 4, and the performance criteria are not respected."
is replaced by: "indicators of Table 4, and the performance criteria defined by Hanna and Chang (2012) and Herring and Huq (2018) are not respected."

*Table 7, caption: Replace 'correspondent' by 'corresponding'*
**Author response**: Done.

*l. 444: 'concentrations' misspelled*
**Author response**: Corrected.

[revised manuscript text omitted]

**3.1  Setup of regional-scale simulations**

The two-way SinG model is applied over Paris city (domain 4), using a spatial resolution of 1 km × 1 km. Initial and boundary conditions are obtained from one-way nesting simulations using Polair3D over three additional simulations covering Europe (domain 1), France (domain 2) and Île-de-France region (domain 3). The spatial resolution for those simulations is 45 km × 45 km, 9 km × 9 km and 3 km × 3 km, respectively. Figure 1 illustrates the different domains, with domain 4 corresponding to the Paris city domain. The four nested simulations over the domains shown in Figure 1 use the same vertical discretization with 14 levels between 0 and 12 km, represented in Figure 2.

[Figure]

[Figure]

**Figure 1.** Regional-scale domains: Europe (domain 1, with a spatial resolution of 45 km × 45 km), France (domain 2,  with a spatial resolution 9 km × 9 km) and Île-de-France region (domain 3, with a spatial resolution 3 km × 3 km) for one-way nesting simulations using Polair3D, and Paris city (domain 4, with a spatial resolution of 1 km × 1 km) for simulations with SinG.

[Figure]

**Figure 2.** Vertical levels used in all regional-scale simulations performed with Polair3D and SinG.

[revised manuscript text omitted]

**7  Conclusions**

In this study, a Street-in-Grid (SinG) multi-scale simulation is performed over Paris city, with a two-way dynamic coupling between the local (street) and regional (background) scales. For Paris, 3819 streets are considered and different databases are used to determine the width and height of each street. A stationary approach may be used to compute pollutant concentrations in the streets, by performing a mass balance between emission, deposition and vertical and horizontal mass transfer. Although this approach is reasonable to estimate $NO_x$ concentrations or the concentration of inert pollutants, it is not appropriate to compute the concentrations of reactive pollutants such as $NO_2$ or NO. A non-stationary dynamic approach was implemented, by solving with a second order numerical scheme the transport of pollutants and chemistry. This approach proved to be numerically stable, with a good agreement between observed and simulated concentrations of $NO_2$ and $NO_x$ at both regional and local scales.

In the streets, $NO_x$ and $NO_2$ concentrations simulated by SinG compare well to measurements performed at traffic stations. For $NO_2$ concentrations, the statistical indicators obtained with SinG and the street model (MUNICH) respect the most strict performance criteria (Hanna and Chang, 2012) at traffic stations. However, NO concentrations are strongly underestimated at traffic stations located in streets that converge in big squares. This underestimation is probably due to the short life time of NO, for which the assumption of uniform concentrations in wide streets and big squares may not be appropriate. At the regional scale, SinG performs also well in simulating $NO_x$ and $NO_2$ concentrations, and the most strict critera are respected at background stations.

The influence of the two-way dynamic coupling between the regional and local scales is assessed by comparing the concentrations simulated with SinG to those simulated with MUNICH. $NO_x$ and $NO_2$ concentrations simulated with SinG and MUNICH are strongly correlated to traffic emissions, and the highest concentrations are observed in the ring road around Paris city ("boulevard périphérique"), where emissions are the highest. Similarly, at both the local and regional scales, the influence of the dynamic coupling is larger in areas where traffic emissions are high. $NO_2$ concentrations simulated with SinG are in general larger than those simulated with MUNICH, especially in high emission areas, because the background concentrations in SinG are influenced by the high $NO_x$ concentrations of the street network. The influence of the two-way coupling depends not only on the emission strength, but also on the aspect ratio (height over width) of the street. Although, on average over the streets of Paris, the influence of the two-way coupling on $NO_2$ concentrations in the street is only 7.5%, it can reach values as high as 63%. The influence of the two-way coupling on background regional $NO_2$ concentrations can be large as well: 11% on average over Paris with a maximum relative difference of 34%. Because NO background concentrations are very low, and because of its short lifetime, NO concentrations are less sensitive to two-way dynamic coupling than $NO_2$.

**Appendix A: Statistical parameters**

**A1 Definitions**

- FB: Fractional bias

$$FB = 2\left(\frac{\bar{o}-\bar{c}}{\bar{o}+\bar{c}}\right)$$

- MG: Geometric mean bias

$$MG = exp(\overline{ln(o)} - \overline{ln(c)})$$

- NMSE: Normalized mean square error

$$NMSE = \frac{\overline{(o-c)^2}}{\overline{oc}}$$

- VG: Geometric variance

$$VG = exp[\overline{(ln(o) - ln(c))^2}]$$

- NAD: Normalised absolute difference

$$NAD = \frac{\overline{|c-o|}}{(\bar{c}+\bar{o})}$$

- FAC2: Fraction of data that satisfy

$$0.5 \le \frac{c}{o} \le 2.0$$

Where $o$ and $c$ represent the observed and simulated concentrations respectively.

**A2 Statistical parameters at all traffic stations**

[revised manuscript text omitted]